# Phenol Biological Metabolites as Food Intake Biomarkers, a Pending Signature for a Complete Understanding of the Beneficial Effects of the Mediterranean Diet

**DOI:** 10.3390/nu13093051

**Published:** 2021-08-31

**Authors:** Juana I. Mosele, Maria-Jose Motilva

**Affiliations:** 1Cátedra de Fisicoquímica, Departamento de Química Analítica y Fisicoquímica, Facultad de Farmacia y Bioquímica, Universidad de Buenos Aires, Buenos Aires C1113AAD, Argentina; jimosse@docente.ffyb.uba.ar; 2CONICET-Universidad de Buenos Aires, Instituto de Bioquímica y Medicina Molecular (IBIMOL), Buenos Aires C1113AAD, Argentina; 3Instituto de Ciencias de La Vid y del Vino (ICVV), Consejo Superior de Investigaciones Científicas (CSIC), Gobierno de La Rioja, Universidad de La Rioja, 26007 Logroño, Spain

**Keywords:** food intake biomarkers, Mediterranean diet, phenol metabolites, metabolomics, polyphenols

## Abstract

The Mediterranean diet (MD) has become a dietary pattern of reference due to its preventive effects against chronic diseases, especially relevant in cardiovascular diseases (CVD). Establishing an objective tool to determine the degree of adherence to the MD is a pending task and deserves consideration. The central axis that distinguishes the MD from other dietary patterns is the choice and modality of food consumption. Identification of intake biomarkers of commonly consumed foods is a key strategy for estimating the degree of adherence to the MD and understanding the protective mechanisms that lead to a positive impact on health. Throughout this review we propose potential candidates to be validated as MD adherence biomarkers, with particular focus on the metabolites derived from the phenolic compounds that are associated with the consumption of typical Mediterranean plant foods. Certain phenolic metabolites are good indicators of the intake of specific foods, but others denote the intake of a wide-range of foods. For this, it is important to emphasise the need to increase the number of dietary interventions with specific foods in order to validate the biomarkers of MD adherence. Moreover, the identification and quantification of food phenolic intake biomarkers encouraging scientific research focuses on the study of the biological mechanisms in which polyphenols are involved.

## 1. Introduction

### 1.1. Food Intake Biomarkers

The term “biomarker” is understood by the Food and Drug Administration [1] (www.fda.gov; accessed on 20 May 2020) as “a defined characteristic that is measured as an indicator of normal biological processes, pathogenic processes, or responses to an exposure or intervention, including therapeutic interventions. Molecular, histologic, radiographic, or physiologic characteristics are types of biomarkers”. The choice of a particular biomarker will be determined according to the aim of the study and its suitability will be reflected in its susceptibility to change. In this review we describe a selective group of food polyphenols and their biological metabolites identified in biofluids (urine, plasma and faeces) for the purpose of considering them as potential biomarkers of adherence to the Mediterranean diet (MD). For this purpose, the study is restricted to the most representative plant foods of the MD considering their frequency of consumption and seasonality.

Food is a complex matrix in which a broad and diverse group of compounds are present in different proportions. Beyond a nutritional need, humans also eat for pleasure. Therefore, they are constantly exposed to diverse types of foods and, consequently, to their components. A food intake biomarker is a molecule that derives from a specific food component (macro- or micro-nutrient) and its occurrence in biofluids is a direct indicator of the degree of exposition (intake) to a particular food [2,3]. Food biomarkers represent the chemical forms in which the original compounds are present in the circulation and reach the target organs or tissues where they carry out the expected metabolic functions [4]. Thus, establishing a more accurate association between the characteristics of a diet and its impact on health is a robust and objective perspective. In addition, the use of suitable intake biomarkers for a specific food also contributes to establishing the degree of compliance in dietary intervention studies.

### 1.2. Food Polyphenols

Most food components are essential for life. Others, although dispensable, accomplish body functions related with protective effects on health. Due to this property, these types of substances are called “bioactive compounds”. Among them, polyphenols have gained special scientific interest. Polyphenols are secondary metabolites of plants that are classified into flavonoids and non-flavonoids. In turn, flavonoids are subdivided into flavan-3-ols, anthocyanins, flavonols, flavanones and isoflavones. The non-flavonoids include phenolic structures with diverse chemical characteristics such as phenolic acids (hydroxycinnamic and hydroxybenzoic acids), stilbenes, phenolic alcohols, and non-hydrolysable and hydrolysable tannins [5].

More than 500 different polyphenols with a heterogeneous pattern of distribution have been described in food [5]. On the one hand, some phenolic compounds are found in a huge variety of foods whereas others are characteristic of a reduced group of foods. In addition, the circulating biological metabolites are not necessarily the same as the polyphenols present in food since the latter undergo different metabolic modifications once absorbed. In consequence, a full knowledge of their metabolism is required to understand the mechanisms of polyphenols in the prevention of chronic diseases.

### 1.3. Polyphenol Metabolome

The study of the metabolic transformation that polyphenols undergo in the human body due to the activity of endogenous and exogenous enzymes is called metabolomics [6,7]. As result, a complete description of their circulating forms and, therefore, how phenol metabolites are present in organs and target tissues, and the form they are excreted is possible [6,8,9].

After ingestion of food, polyphenols must necessarily cross several biological barriers before reaching the target organ or tissues or being excreted. Digestion in the upper part of the digestive tube includes oral, gastric and intestinal steps in which endogenous enzymes participate. The small intestine is where the major absorption takes place and, therefore, the bioaccessibility and bioavailability of the original compounds are defined [10]. Once absorbed, phenolic compounds undergo phase I and II metabolism in the intestine and liver leading to the generation of a complex combination of different patterns of sulphated, glucuronidated and methylated conjugated metabolites [11,12,13]. Most of these metabolites are excreted in urine and a fraction is recirculated into the colon where they are exposed to further catabolism by gut microbiotaHowever, it is documented that most food polyphenols are poorly absorbed in the small intestine and a relevant percentage reaches the colon. In the gut, parent compounds are modified by microbial metabolism, including dehydroxylation, demethylation, ring fission and decarboxylation, among others [10,14,15]. In addition, colonic metabolites can be absorbed by colonocytes and reach organs and tissues and are excreted via urine [12].

Thus, the original compounds in foods are transformed in the human body into related compounds with similar chemical structures but not necessarily the same activity. These diverse modifications of parent compounds into related phase I and II metabolites constitute the so-called “food metabolome” [16,17]. The identification of these metabolites in biofluids has been associated with the exposure to a particular food while their quantification indicates the degree of exposure [4]. The kinetics of intake biomarkers, including postprandial absorption, metabolism and half-life of circulating compounds, is normally studied after an acute intake of a polyphenol-rich food or extract [4,11,18]. Medium- and long-term dietetic interventions, as well as epidemiological studies, are designed to determine the “steady state” of phenolic metabolites due to the sustained intake of one or more phenolic-rich foods [3]. As biochemical mechanisms of polyphenols are linked to disease prevention, there is a need to know the cycle of polyphenols in the body clearly in order to understand their biological relevance.

## 2. Phenol Metabolites as Biomarkers of Adherence to Mediterranean Diet: Is It Possible?

According with the Mediterranean Diet Foundation [19], “The MD is a valuable cultural heritage that is much more than a dietary pattern. It encompasses a balanced lifestyle including recipes, cooking methods, celebrations, particular habits, local products and cultural habits”. The ten principles of the MD are: (1) use olive oil as your main fat source; (2) eat plenty of plant products such as fruits, vegetables, legumes and nuts; (3) bread and other grain products (pasta, rice, and whole grains) should be a part of the everyday diet; (4) fresh and locally low-processed products are preferred; (5) consume dairy products on a daily basis, mainly yogurt and cheese; (6) red meat should be consumed in moderation and if possible as a part of stews and other recipes; (7) consume fish abundantly and eggs in moderation; (8) fresh fruit should be your everyday dessert and, sweets, cakes and dairy desserts should be consumed only on occasion; (9) water I preferred over other beverages and wine is consumed in moderation at meal times; (10) be physically active every day. The new pyramid imitates previous models: those foods at the base should support the diet whereas those at the top should be eaten in moderate/low amounts. It is not just about prioritizing some food groups over others, but also paying attention to the selection, way of cooking and eating. According to this model, plant products represent the core of the daily intake, contributing with large variety of fibres and essential micronutrients. Moreover, social and cultural elements characteristic of the Mediterranean way of life have been incorporated into the graphic design [19].

The MD can be interpreted as a dynamic diet since there is a marked seasonal variability, particularly evident in the intake of fruit and vegetables. This trend in the consumption of seasonal products is associated with the high value that Mediterranean inhabitants attribute to local products [20]. The preference for fresh products is partly due to the proximity of the production zones for these high nutritional quality food products. This elevated quality is linked to their polyphenol content to which many authors have attributed the protective effects of the MD [21].

Is it possible to elaborate an agreed and unified list of biomarkers associated with MD consumption? An ideal biomarker of adherence to MD should respond to the intake of traditional polyphenol-rich plant-foods. So, we propose polyphenols and their related metabolites as biomarkers for adherence to MD. This choice takes into consideration two main aspects: (i) the changes observed in the phenolic metabolome of biofluids after the intake of phenol-rich foods typical of the MD, and (ii) the presence in biofluids of phenol biological metabolites could be associated with modifications in the metabolic functions or activities in the body whose effects are closely related to protection against chronic diseases. Based on these criteria, the main objective of this review was carried out a descriptive bibliometric analysis on phenolic biological metabolites detected in biofluids associated with the intake of the most representative foods of the Mediterranean diet. For this, we proposed to analyse the available research activity between 2000 and 2020 dedicated to the identification of phenolic metabolites in human biofluids. A systematic search was done in PubMed database and scientific literature was selected and evaluated for their inclusion in this review. Inclusion criteria used to discriminate most relevant studies were authors productivity, relevance of authors in the field, most productive countries, relevance of food included, most relevant keywords, methodology used to identify phenolic metabolites and human diet intervention studies. We have delimited the period of published data between 2000 and 2020 coinciding with the more productive data generation, the increased interest of various research groups in identify phenolic metabolites and a great advance in technological development in the field of analytical chemistry. Exceptionality, and due to the lack of more current data, some papers published before 2000 were included.

Reliable dietary assessment methods are crucial when attempting to understand the links between diet and chronic disease. Different methods have been used in nutritional epidemiology to estimate food intake including 24 h recalls, weighted food diaries and food frequency questionnaires [22]. However, further progress is required to overcome certain issues such as the effect of subjectivity, correct incorporation of subpopulations and translation into public health messages. For this proposal, nutritional habits would include the specification of different items, such as portion size, frequency of consumption, food composition and daily variations in intake, between others. However, documenting all these individual parameters in large epidemiological studies is not always feasible. Thus, errors in the assessment of the food consumed are to be expected, and an objective and independent validation against quantitatively measurable parameters is needed.

Recently, ‘Systems Epidemiology’ has combined traditional epidemiological methods with such modern high-throughput technologies as genomics, transcriptomics, proteomics, and metabolomics to enhance biological understanding of metabolic pathways in humans [7]. Nutritional epidemiological studies are traditionally based on self-reported dietary assessment methods (24 h recall, food-frequency questionnaires). Despite the extensive and approved use of these tools, they have well-known limitations that do not allow further advances in the human nutrition field, because foods are mixtures of known and unknown constituents and objective biomarkers do not exist for everyone [16]. The use of dietary biomarkers in nutritional epidemiological studies may better capture exposure and improve the level at which diet-disease associations can be established and explored [23]. Over the past decade, nutritional epidemiology has incorporated metabolomics as a promising technique to measure the metabolic products of foods and might therefore identify objective dietary biomarkers which reflect true food exposure.

Two different fractions of the human metabolome are influenced by the diet: the endogenous metabolome and the food metabolome [17]. The former includes all metabolites from the host, which in turn might be modulated by the diet affecting human health. The latter has been defined as the sum of all metabolites directly derived from the digestion of foods, their absorption in the gut, and their biotransformation by the host tissues and the intestinal microbiota [24].

Although some biomarkers of key plant-foods in the MD, such as vegetables, fruits, virgin olive oil or red wine, have been individually described in different human interventional studies, the food metabolome of the MD has yet to be defined. Thereby, the integration of metabolic profiling with reported dietary assessment can be combined to discover biomarkers of food exposure and can help disentangle the molecular mechanisms by which MD affects health and disease.

## 3. Potential Mediterranean Diet Biomarkers: Phenol Biological Metabolites from Recommended Foods on a Daily Basis

### 3.1. Vegetables

#### 3.1.1. Artichokes

Despite the interest in the phenolic composition of this green leaf, there is very little information about the metabolomics of its most representative polyphenols. The principal phenolic compounds in this edible plant belong to the phenolic acid group including di- and mono-hydrocaffeoylquinic, coumaroylquinic, feruloylquinic and caffeic acids, and the flavones luteolin and apigenin glycosides [25,26,27]. The kinetic of the appearance of metabolites in plasma after the acute intake of cooked artichoke behaves in a biphasic way [25]. The first phase occurred between 1–2 h after intake where the maximum concentrations of chlorogenic, caffeic and ferulic acids were observed. The second relevant increase in phenolic species in plasma was detected 8 h after intake and corresponded to the microbial metabolites ferulic, 3-(3′,4′-dihydroxyphenyl)propionic and ferulic and dihydroferulic acids (Table 1). The acute intake of artichoke extract produces a significant increase in urinary ferulic, isoferulic, vanillic and dihydroferulic acids [27]. Although it seems viable to link ferulic and dihydroferulic acids with artichoke intake, more studies are needed to confirm this, specially to evaluate the weight of artichoke intake in the production of this phenolic metabolites considering other foods from MD than can also produce them.

#### 3.1.2. Lettuce

Quantitative analysis of green lettuce has revealed that hydroxycinnamic acids related compounds such as feruloyl quinic acid and caffeoyl feruloyl quinic derivatives represent the main phenolic fraction followed by quercetin glucosides and kaempferol derivatives [28] whereas cyanidin content increase according with the red intensity of the leaf [29]. Despite the high interest in studying the phenolic composition of different varieties of lettuce, as well as its association with health benefits, there are no studies in which the biomarkers of lettuce intake have been investigated. This opens an interesting research field [28,29].

#### 3.1.3. Onions

Onions are used as an ingredient in many traditional MD dishes and consumed raw in salads and other cold preparations. Their contribution to the diet is high considering that they are often consumed in many culinary preparations. Onions are a rich source of quercetin. This is mainly found as glycoside conjugate, especially 4′-*O*-glucoside and 3,4’-diglucoside [11,30,31]. As it is a quercetin rich-food, many studies have been carried out to understand the pharmacokinetic and metabolism of this flavanol. Followed the intake of different onion preparations, a common phenol spectrum was observed in the plasma. This included quercetin glucuronide, quercetin diglucuronide, methylquercetin glucuronide, quercetin glucuronide sulphate and quercetin sulphate as the major metabolites detected (Table 1) [11,13,30,31]. The phenolic profile of the urinary excretion of onion phenolic metabolites is similar to that observed in the plasma. The main metabolites detected were methyl quercetin glucuronide, quercetin glucoside sulphate, quercetin glucuronide, quercetin diglucuronide and quercetin glucuronide sulphate [31]. Despite high variability in terms of metabolites concentrations between volunteers was observed, the phenolic profile was similar [11,30,31,32]. Based on the latter results, quercetin related compounds could be proposed as biomarkers of onion intake. Therefore, after revising the data in the literature, we have noted that there is not much information about the analysis of microbial catabolites of onion polyphenols in biofluids.

#### 3.1.4. Spinach

A study carried out by Passon et al. [33] revealed the main forms of phenolic metabolites associated with spinach intake. Five subjects consumed 5 g ^13^C intrinsically labelled lyophilized and powdered spinach containing 160 µmol of methoxy flavonols from which 70 µmol corresponded to 5,3′,4′-trihydroxy-3-methoxy-6,7-methylendioxyflavone-4′-glucuronide (TMM-4′-glucuronide). The major metabolites detected in the plasma were identified as TMM-glucuronide and TMM-sulphate and these reached their maximum concentrations 7 h after ingestion (Table 1). This indicates colonic metabolism and/or enterohepatic recirculation. In addition, spinacetin and patuletin were identified (but not quantified) in human plasma for the first time after spinach intake [33].

#### 3.1.5. Tomato

Fresh tomato is consumed raw in salads and cold soups or cooked in many dishes in the Mediterranean gastronomy. The main phenolic compounds found in different varieties of tomatoes are homovanillic acid, caffeic acid, ferulic acid, caffeoylquinic acid, naringenin and quercetin 3-rutinoside (rutin) [12,18,34]. Studies carried out to elucidate the metabolic profile of biofluids after the intake of tomato (Table 1) have observed that there is no qualitative difference between raw or cooked tomato, but a slight difference was observed in the quantitative profile [18]. After analysing the data presented by different authors, a common phenolic spectrum was observed in plasma and urine. The phenolic compounds that increased in plasma after tomato intake included naringenin, naringenin glucuronide, ferulic acid glucuronide, isoferulic acid, ferulic acid sulphate and quercetin [12,18,34]. In the case of the urine excretion, 31 different metabolites derived from parent compounds in food were identified. These included free, glucuronidated and sulphated caffeic, coumaric, ferulic hydroferulic, hydroxyphenylacetic, di- and mono-hydroxyphenyl propionic, phenylacetic acids, and non-conjugated hippuric, hydroxybenzoic and homovanillic acids [12,34].

### 3.2. Cereals and Grain Based Products

#### Wheat

In some countries, such as Italy, wheat is widely consumed in the form of pasta and in African Mediterranean countries, cous-cous is the main form of wheat intake. Refined grains contain practically no polyphenols because these compounds are mainly located in the external parts of the grain. However, whole grains products are the preferred form of everyday intake in the MD. In whole wheat, 34 phenolic compounds (including isomer forms) belonging to the phenolic acid, flavonoid, stilbene, proanthocyanidin and lignin chemical classes have been described [35]. In recent years, much attention has been focused on the phenolic lipids named alkylresorcinols. These are 1,3-dihydroxybenzene derivatives with an odd-numbered alkyl chain at position 5 of the benzene ring [36]. Based on the consistent results observed in various intervention and observational studies, both alkylresorcinols and their metabolites have been proposed as strong candidates for biomarkers of whole-wheat intake [36,37]. These compounds (Table 2) were identified in plasma and urine as 3,5-dihydroxy benzoic acid (3,5-DHBA), 3,5-dihydroxyphenil propionic acid (3,5-DHPPA), 5-(3,5-dihydroxyphenyl)pentanoic acid (DHPPTA), 2-(3,5-dihydroxy)hippuric acid, dihydroxycinnamic acid and 3,5-dihydroxycinnamic acid amide (DHCA-amide) [36,37]. The DHBA, DHPPA and DHPPTA metabolites were also detected in urine as glucuronide or sulphate conjugates [36]. These metabolites have also been shown to correlate well with self-reported food records, which means they are suitable medium- to long-term biomarkers of whole rye, oat and wheat intake [37].

In addition, other metabolites of whole grain polyphenols can reinforce the estimation of whole wheat intake. In a study in which a portion of refined wheat was replaced by whole wheat in the diet of 80 healthy overweight/obese subjects, with low intake of fruit and vegetables and sedentary lifestyle, an increase in the dihydroferulic acid concentration in the plasma was observed as was a rise in ferulic acid in the urine and faeces [38].

### 3.3. Virgin Olive Oil

Probably, the use of virgin olive oil (VOO) as leading fat source is one of the main dietary features of the MD. VOO differs from other fats as it contains a large percentage of monounsaturated fatty acids, main oleic acid, and a large spectrum of bioactive compounds. Regarding the health claims which may be made about foods (European Regulation EU 432/2012) [39], two fractions of VOO can be considered, namely oleic acid and polyphenols. For oleic acid, the authorized claim is “Replacing saturated fats in the diet with unsaturated fats contributes to the maintenance of normal blood cholesterol levels. Oleic acid is an unsaturated fat”. The claim may be used only for food which is high in unsaturated fatty acids (Regulation (EC) No 1924/2006) [39]. Oleic acid is the main monounsaturated fatty acid (MUFA) in olive oil (70–80%). However, among the selective consumption biomarkers found in olive oil, oleic acid is the main part of the fatty acid composition of such different vegetable oils as rapeseed oil (average 72%), peanut oil (average 71%), almond oil (average 68%), rice bran oil (average 44%) and sesame oil (average 42%) [40]. Consequently, oleic acid and its possible biological metabolites are not good candidates as selective biomarkers of VOO consumption. By contrast, some specific components of the VOO phenolic fraction could be better candidates as intake biomarkers.

The phenolic compounds present in various parts of *Olea europaea* L. belong to six main families: phenolic acids, phenolic alcohols, flavonoids (flavones, flavonols, flavanones, and flavanols), isochromans, lignans, and secoiridoids and their derivatives [41]. Secoiridoids are rare in plant species and occur mainly in Oleaceae species, being abundant in O. europaea (leaves and fruit). During VOO extraction, the secoiridoids oleuropein and ligstroside from olive fruit are enzymatically hydrolysed, forming their decarboxymethyl dialdehydic forms of oleuropein (the dialdehydic form of elenolic acid linked to hydroxytyrosol or 3,4-DHPEA-EDA) and ligstroside aglycons (the dialdehydic form of elenolic acid linked to tyrosol or *p*-HPEA-EDA), respectively, and their aldehydic forms of oleuropein (the aldehydic form of elenolic acid linked to hydroxytyrosol or 3,4-DHPEA-EA) and ligstroside (the aldehydic form of elenolic acid linked to tyrosol p-HPEA-EA), respectively. These secoiridoids are the most abundant polyphenols in VOO. In 2012, the EFSA (European Food Safety Authority) [42] recognized a health claim associated with the contribution of “olive oil polyphenols” for protecting blood lipids from oxidative stress. This is only allowed for “olive oils containing at least 5 mg of hydroxytyrosol and its derivatives (e.g., oleuropein complex and tyrosol) per 20 g of olive oil” (Commission Regulation (EU) 432/2012; EFSA 2012) [39,42]. Oleuropein aglycon derivatives (dialdehyde and aldehyde forms) are the main contributors to the health claim. As a consequence of the approved health claim, the majority of research has focused on the bioavailability studies of the olive oil polyphenols in the analysis of the absorption and excretion of hydroxytyrosol metabolites. During gastrointestinal digestion, secoiridoids are hydrolysed to form hydroxytyrosol and tyrosol and which are subsequently extensively metabolized in the gut and liver and so are mainly found in biological fluids as phase-II metabolites [4,43]. This extensive conjugation and further rapid excretion could also explain the low plasma concentration of free hydroxytyrosol [4].

Different human intervention studies that target the bioavailability of olive oil phenols have detected as the most common metabolites in plasma and urine, the phase-II metabolites of hydroxytyrosol. In a randomised, cross-over acute intake human study [4] with a dose of 30 mL of three phenol-enriched virgin olive oils with a phenolic content of 250, 500, and 750 mg total phenols/kg oil, hydroxytyrosol sulphate and hydroxytyrosol acetate sulphate were selected as the main biological metabolites of hydroxytyrosol after olive oil ingestion (Table 3). The plasma pharmacokinetics of these metabolites showed a dose-dependent response. However, a completely linear response was not observed after the intake of the phenol enriched oils with 750 and 500 mg phenols/kg, respectively. This could be explained by the saturation of the transport systems involved in the intestinal efflux and/or phase-II metabolism enzymes. These biomarkers of olive oil phenols were confirmed in a randomized, double-blind, controlled, cross-over trial, in which 33 hypercholesteraemic subjects received 25 mL/day of a VOO enriched with its own phenolics (500 mg phenols/kg oil; FVOO) and a VOO enriched with its own phenolics plus complementary phenolics from thyme (500 mg phenols/kg oil, 50% from olive oil and 50% from thyme respectively; FVOOT) for 3 weeks [44]. Based on the analysis of plasma and urine samples, before and after each intervention period (3 weeks), the authors proposed hydroxytyrosol sulphate and hydroxytyrosol acetate sulphate as suitable biomarkers for monitoring compliance with VOO intake as their values in plasma or/and 24 h urine were significantly higher after FVOO than in baseline pre-intervention concentrations. The viability of these hydroxytyrosol metabolites as biomarkers of olive oil phenol intake was reinforced by the significant positive correlation observed between the number of bottles consumed (25 mL oil/day) during the intervention period (3 weeks), and the phenol metabolite concentration detected in 24 h urine samples for each volunteer. In contrast, the plasma concentration of hydroxytyrosol metabolites showed no significant positive correlations. When the authors compared the urinary levels of hydroxytyrosol metabolites after FVOO and FVOOT consumption, it was noteworthy that although FVOOT contained half the amount of hydroxytyrosol derivatives, the concentrations excreted were higher than expected (Table 3) [44]. This indicates a protective effect of thyme phenols during digestion which resulted in greater efficacy in the bioavailability of hydroxytyrosol. This observation reinforces the interest of possible interactions between different diet phenols.

Similar biomarkers of VOO intake after the acute intake of 40 mL of VOO with high content in phenolic compounds (400 mg/L) were proposed by Orozco-Solano et al. [45]. In this study, the metabolites with the highest concentration in plasma were hydroxytyrosol monosulphate and monoglucuronide derivatives. Note the quantification of free hydroxytyrosol in human plasma. Other studies propose a complex metabolism of secoiridoids that includes phase-I (oxidation, hydrogenation, hydration, decarboxylation, hydroxylation, and methylation) and phase-II reactions (conjugation reactions and including glucuronidation, sulphoconjugation, acetylation, and glutamination) [14,43,46]. In a study by García-Villalba et al. (2010) [41], such metabolites as the hydroxylated and methylated forms of hydroxytyrosol and oleuropein aglycone were found in urine after the acute intake of 50 mL of VOO (Table 3). A recent study by Silva et al. [46] confirmed the complexity of the metabolism of secoiridoids after the intake of a single dose of 50 mL of VOO (equivalent to 6 mg of hydroxytyrosol and its derivatives). Phase-I metabolites involved in hydroxylation and hydration, such as *p*-HPEA-EDA + H_2_O and 3,4-DHPEA-EA + OH, were detected in plasma. In urine, 3,4-DHPEA-EA + H_2_ + glucuronide and methyl 3,4-DHPEA-EA + H_2_ + glucuronide were selected as biomarkers of intake.

As well as being found in plasma and urine samples, hydroxytyrosol monoglucuronide, hydroxytyrosol monosulphate, tyrosol glucuronide, tyrosol sulphate and homovanillic acid sulphate have been detected in low-density lipoprotein (LDL cholesterol) obtained 60 min post-intake of 50 mL of VOO [47]. Regarding HDL cholesterol, hydroxytyrosol sulphate, homovanillic acid sulphate and homovanillic acid glucuronide were detected following dietary supplementation with VOO (366 mg phenols/kg oil) for 3 weeks (a daily dose of 25 mL) [48]. The concentration of hydroxytyrosol sulphate in HDL (49.48 ng/mgAPO-A1) was higher than the level observed in LDL samples (34.22 ng/mg APO-B) [47]. Based on the results of these numerous studies about the metabolism of VOO polyphenols, hydroxytyrosol sulphate is the most common metabolite detected in the biological samples obtained after the intake of different doses of VOO, different phenolic concentrations in the oil and different intake patterns (acute or sustained). In consequence, hydroxytyrosol sulphate could be considered a good biomarker of VOO intake analysed in such different biological samples as cholesterol particles (HDL and LDL), plasma and urine.

Regarding colonic metabolites, after the sustained intake (3 weeks) of a daily dose of 25 mL of a phenol-enriched olive oil (500 mg phenols/kg oil), an increase in the concentration of phenylacetic and phenylpropionic acids was observed in human faeces [14]. Of special interest was the significant increase in the concentration of free hydroxytyrosol after the phenol-rich olive oil diet supplementation period. In this way, modulating the concentration of phenol colonic metabolites through the diet could have an impact on human intestinal health.

### 3.4. Fresh Fruits

#### 3.4.1. Apples

Apple provides the highest percentage of flavonoid intake among fruits in the MD population [49]. Red-peel apple cultivars are an important source of different polyphenol subclasses including anthocyanins (cyanidin galactoside being the most abundant), phenolic acids, flavan-3-ols (epicatechin and its polymerized forms), dihydrochalcones, flavonols (quercetin derivatives), and flavones and flavanones [50]. Many studies have selected juice or cider as the apple source when studying the metabolome of apple polyphenols. Kahle et al. [51] analysed the phenolic profile of blood and urine collected from five healthy volunteers after the intake of cloudy apple juice. Most of the polyphenols from apple juice were detected in the serum and urine in their conjugated forms. The main metabolites quantified in serum, after glucuronidase/sulphatase enzymatic treatment, were caffeoylquinic acid, caffeic acid, 4-*p*-coumaroylquinic acid, (−)-epicatechin, phloretin, and quercetin (Table 4). Only caffeic acid in its free form was detected in urine samples, whereas after hydrolysis by glucuronidase and sulphatase, the free forms of quercetin, caffeic acid, chlorogenic acid, 4-*p*-coumaroylquinic acid, phloretin, and (−)-epigallocatechin were quantified. Similarly, after an acute intake of 250 mL of natural cloudy apple juice [52], most of the phenolic metabolites detected in plasma and urine were the glucuronide, methyl and sulphate conjugates of apple polyphenols. During digestion, the chalcone phlorizin, the most characteristic polyphenol of the Rosaceae family and therefore of apples, is first deglycosylated to give phloretin, which then undergoes phase-I and -II metabolism. In addition, only small amounts of chlorogenic acids were detected in urine and no (epi)catechins, quercetin or phloridzin were found [52] (Table 4). In a study by Saenger et al. [53], phloretin, epicatechin, and procyanidin B2 were used as short-term urinary intake biomarkers for a period of at least 12 h after consumption of 200, 400 and 790 g of apple quavers, corresponding to one, two, or four apples, respectively. In this dose-response study, the maximum urine concentration of phloretin and epicatechin for each consumption group was reached after 3 h (±60 min), and of procyanidin B2 after 3–6 h of the apple intake. In a recent study by Yuste et al. [9], a total of 37 phase-II and microbial phenolic metabolites were detected in plasma and urine after an acute intake of 80 g of freeze-dried red-fleshed apple snack. Among all the metabolites generated, phloretin glucuronide was proposed as the best candidate as a biomarker of apple intake. Additionally, cyanidin-3-*O*-galactoside and peonidin-3-*O*-galactoside were proposed as biomarkers of the intake of red-fleshed and red-skinned apples. However, these anthocyanin metabolites could be biomarkers of the intake of other important red fruit sources of anthocyanidins, such as chokeberries and lingonberries. Based on all the studies analysed, the phloretin biological metabolites (glucuronide and sulphate conjugates), analyzed preferably in urine, could be proposed as biomarkers of apple intake based on the specificity of their precursor phloridzin as phenolic compound characteristic for apples. Regarding phenol colonic metabolites in plasma during postprandial period, especially those late time points (12 h after food intake), these are normally missing in studies when blood sampling is done by venepuncture because blood sampling is usually only possible between 0 and 6–8 h. In a recent study by Yuste et al. [54], a method based on the use of Dried Blood Spot (DBS) cards (blood sampling) combined with liquid chromatography coupled to mass spectrometry (UPLC-MS/MS) was optimised and applied for determining phenolic metabolites in human blood and plasma samples after the acute intake of an apple snack. Similar to other studies, the phloretin glucuronide detected in blood and plasma samples was selected as a biomarker of apple intake. Interestingly, dihydroxyphenylpropionic acid sulphate, (methyl)catechol sulphate, catechol glucuronide, and hydroxyphenyl-γ-valerolactone glucuronide were quantified in blood samples. They reached their maximum concentrations 12 h after consumption of the apple snack, indicating that they could be products of the gut microbiota catabolism in the colon. So, self-sampling of blood by the volunteers using DBS cards opens new possibilities for the identification of the circulating phenol metabolites which can be used as biomarkers for the consumption of specific foods, since it allows simple and frequent blood sampling over 24 h.

#### 3.4.2. Pear

Pear is another common fruit whose production and consumption is growing in all temperate regions in more than 50 countries around the world due to its flavour and manifold cultivars [55]. The polyphenolic fraction of pears contains mainly flavonoids and phenolic acids [56]. In the flavonoid subgroup of flavan-3-ols, catechin, epicatechin and epicatechin gallate are the most abundant, as are kaempferol, kaempferol-3-glucoside, quercetin and quercetin-3-glucoside among the flavonols whereas phenolic acids arerepresented basically by gallic and chlorogenic acids [54]. Unlike with apple polyphenols, the human metabolism of those found in pears has scarcely been studied. Among the exceptions, the study carried out by Nieman et al. [57] deserves mention. In this, the phenolic metabolites in plasma were analysed in male athletes after cycling 75 km while consuming water or pears equivalent to 0.6 g carbohydrate/kg each hour. Compared with the consumption of only water, an increase of ferulic, dihydroferulic, 3-(4-hydroxyphenyl) propionic, 4-hydroxy hippuric and hippuric acids was observed (Table 4).

#### 3.4.3. Oranges

Oranges are consumed as whole fruit or in the form of juice. The flavanones hesperetin and naringenin in their 7-*O*-rutinoside derivatives, respectively named hesperidin and nariutin, respectively, are the most abundant flavonoids in orange juice. Other polyphenols are eriodictyol rutinoside, apigenin diglucoside, methylnaringenin rutinoside and glucosides of ferulic and coumaric acids [58,59,60]. Studies concerning acute ingestion of orange juice have given an understanding of many aspects of the human metabolism of citrus polyphenols (Table 4). In line with the juice composition, the collection of phenolic metabolites corresponds with their concentration in the juice, hesperetin glucuronides and sulphate-glucuronides being higher than the same derivatives of naringenin [58]. A total of 19 flavanone metabolites and 65 phenolic microbial metabolites were reported in urine after the consumption of orange juice [58,59,60]. Ten hesperetin, seven naringenin and two eriodictyol metabolites were described in the first group. Hesperetin-*O*-glucuronides were the main metabolites detected followed by naringenin-*O*-glucuronides and hesperetin-3′-*O*-sulphate. The group of microbial metabolites was composed of 14 phenylpropanoid acid derivatives, 20 phenylpropionic acid derivatives, 12 phenylacetic acid derivatives, 10 benzoic acid derivatives, 2 hydroxycarboxilic acid derivatives, 3 benzotriol derivatives and 4 benzoylglycine derivatives. From this category 3-(3′-hydroxy-4′-methoxyphenyl)hydracrylic acid, 3-(3′-hydroxy-4′-methoxyphenyl)propionic acid (dihydroisoferulic acid), 3-(3′-methoxy-4′-hydroxyphenyl)propionic acid (dihydroferulic acid) and 3′-hydroxyhippuric acid, appeared in urine free or as phase II metabolites after orange juice consumption but not after intake of the placebo drink [59,60]. Among all these metabolites, special attention should be paid to hesperetin-*O*-glucuronides and the microbial breakdown product of hesperitin, 3-(3′-hydroxy-4′-methoxyphenyl)hydracrylic acid, since they could be proposed as biomarkers of orange juice consumption. The reason could be that these phenolic metabolites are excreted in substantial amounts and were absent from urine collected after the consumption of the placebo drink and also their concentrations circulating in plasma are sensitive to the dose of juice ingested [15,58,59,60].

#### 3.4.4. Pomegranate

Pomegranate fruits are mostly consumed during autumn and winter. In recent years, both 100% pure and blended pomegranate juices (mixed with grapes, oranges, vegetables), have attracted the interest of consumers due to the promotion of their antioxidant properties. Ellagitannins, punicalagin and punicalin are the main phenolic compounds in pomegranate juice but it also contains variable amounts of ellagic acid, anthocyanins and, to a lesser extent, flavan-3-ols [10,61]. Several human acute intake or medium-term dietary interventions have been carried out in order to study the metabolism of pomegranate polyphenols, most of them using juice [8,61,62] or phenolic extracts obtained from pomegranate by-products [62,63,64]. Considering their abundance and the fact that they are found in a few selected foods, ellagitannins could be ideal biomarkers of pomegranate intake, however, they are poorly absorbed and further metabolized by intestinal microbiota. Hence, urolithins, their microbial metabolites, can be proposed as candidate markers for pomegranate intake. Ellagic acid, either contained in the fruit or originating from intestinal hydrolysis of ellagitannins, is further catabolized by intestinal microbiota resulting in the generation of pentahydroxylated urolithins. These then rapidly undergo successive dihydroxylations [10]. Several studies have detected urolithins, especially monohydroxylate urolithin glucuronide (urolithin A glucuronide) as the main microbial metabolite of ellagitannins [8,61,62,64]. The individual variability in the quantification of urolithin A in biofluids allows different phenotypes or urolithin producers to be distinguished, these being high urolithin producers (>5 µM), low urolithin producers (<5 µM) and very low urolithin producers (undetectable) [8]. Consequently, the selection of urolithins as biomarkers of the consumption of pomegranate or its derivatives is conditioned by the individual profile of the intestinal microbiota, or metabotype.

#### 3.4.5. Grapes

Due to their phenolic content, we will focus on the analysis of potential biomarkers of red grape intake. The phenolic composition of red grapes includes flavan-3-ols (epicatechin, catechin, gallocatechin, epigallocatechin, and epicatechin-3-*O*-gallate), diverse polymeric forms of procyanidins, anthocyanins in their glucoside and acetylated conjugates and flavonols (normally as *O*-glycosides of myricetin, luteolin, quercetin, kaempferol and isorhamnetin). In the group of non-flavonoids, the presence of hydroxycinnamic and hidroxybenzoic acids, tartaric esters of hidroxycinnamic acids and the stilbene resveratrol have been reported [65,66,67]. The intake biomarkers of red grapes have been investigated in humans after the acute intake of grape juice and pomace extract [6,65,66,67]. Considering the amount of metabolites detected in plasma, the absorption of the intact forms of red grape polyphenols is poor and a significant proportion reaches the colon [67]. It could be expected that, according to the high amounts of malvidin, delphinidin and cyanidin derivatives in the grapefruit, anthocyanins could be considered as good candidates as biomarkers of red grape consumption. Although detected in plasma and urine, mainly in their 3-glucoside and *O*-glucuronide forms, anthocyanins are poorly absorbed, rapidly excreted and are absent in the biofluids of some volunteers [65,66,67]. On the other hand, resveratrol has been considered a good grape intake biomarker despite its limited absorption [68].

In early studies, when the microbial metabolites of phenolic compounds were relatively unexplored, authors focused on the identification of the native forms of the phenolic compounds present in the fruit [65]. Over the years, the identification of microbial phenolic catabolites has earned the interest of researchers. More than 40 metabolites have been determined in plasma and urine after the acute intake of 350 mL of Concord grape juice. Of these, 8 are of particular interest due to their high concentration in urine. These are epicatechin sulphate, *O*-methyl-epicatechin-*O*-sulphate, m-dihydrocoumaric acid, dihydrocoumaric acid-*O*-sulphate, dihydroxyphenyl propionic acid-3′-*O*-sulphate, ferulic acid sulphate, dihydroferulic acid, dihydroferulic acid 4′-*O*-sulphate and isoferulic acid-3′-*O*-glucuronide (Table 4) [66,67]. In addition, it was observed that the intake of 250 mL of an aqueous drink based on a grape pomace extract produced a considerable increase of epicatechin-sulphate in plasma and urine, together with such other potential microbial metabolites as methylpyrogallol sulphate, hydroxyphenyl propionic acid-sulphate, and hydroxyphenyl-γ-valerolactone-sulphate isomers [6]. In summary, the microbial metabolites formed by the colonic catabolism of proanthocyanidins, such as hydroxyphenyl-γ-valerolactone-sulphate isomers, could be considered as red grape intake biomarkers, nonetheless procyanidins are widespread in nature.

### 3.5. Legumes

Once of the representative characteristics of the MD is the high intake of dried legumes in the regular diet, especially important due to their presence in many typical dishes are chickpeas and lentils. Despite the high interest in the characterization of the phenolic composition of pulses, there are no works focused on the study of the phenolic metabolites associated with their consumption. The phenolic composition of chickpeas depends on the variety. Phenolic acids, including *p*-hydroxybenzoic, syringic and gentisic acids, are the major phenolic compounds in the legumes. Although in lesser amounts, some flavonoids, such as catechin, epicatechin, luteolin-8-*O*-glucoside, myricetin-3-*O*-rhamnoside, quercetin-3-*O*-galactoside and quercetin-3-*O*-rhamnoside, are also present in chickpeas [70,71]. Regarding lentils, the main phenolic compounds are epicatechin, catechin and phenolic acids, such as syringic and coumaric acids [72].

### 3.6. Aromatic Herbs

The use of such aromatic herbs as parsley, basil, thyme, oregano is very widespread in the Mediterranean cuisine. These aromatic herbs are usually consumed fresh as ingredients of several dishes. Regarded their phenolic composition, parsley is rich in apigenin-7-apiosylglucuronide (apiin), apigenin-7-manoylapiosylglucuronide, diosmetin-7-apiosylglucuronide and diosmetin-7-manoylapiosyl glucuronide; basil is rich in quercetin-3-rutinoside and rosmarinic acid; oregano has high amounts of phenolic acids, mainly rosmarinic acid, and flavonoids such as luteolin and apigenin glucosides; and the phenolic composition of thyme is characterized by luteolin glucuronide, rosmarinic acid, and the volatile monoterpenes, thymol and carvacrol [44,73,74].

Human studies in which aromatic herb phenolic metabolites were studied in biofluids are very scarce (Table 5). In the case of parsley, 20 g of daily intake with meals led to a significant urinary excretion of apigenin, but no other metabolites were described [75]. After the intake of an oregano extract, the urinary excretion (48 h) of microbial phenol metabolites, such as 4-hydroxybenzoic, 3-hydroxyphenylacetic, vanillic and ferulic acids, increased [73]. Although no rosmarinic acid and derivate compounds were described in the study of Nurmi et al. [73], glucuronidated and sulphated conjugates of rosmarinic acid and methyl rosmarinic acid were identified in the plasma and urine of volunteers after an acute intake of perlilla extract [76]. Rubió et al. [44] observed an increase in the urinary and plasma concentrations of hydroxyphenyl propionic acid, caffeic acid and thymol sulphate after the dietary supplementation (3 weeks) with 22 g of a phenol-enriched olive oil with a combination of thyme and olive extracts, the authors observe an increase in the urinary and plasma concentrations of hydroxyphenyl propionic acid, caffeic acid and thymol sulphate. These phenolic metabolites were proposed in this study as biomarkers of the intake of thyme phenols.

## 4. Phenol Biological Metabolites from Recommended Foods to Be Eaten in Moderate Amounts

### 4.1. Nuts: Walnuts, Hazelnuts, Almonds

In Mediterranean gastronomy, nuts are typically ingredients of sweets and recently their consumption is in the form of snacks. Most of the available data in the literature refers to human intervention studies in which volunteers consumed mixed nuts and not only a single class [8,77,78]. Each type of nut has a unique phenolic profile. Walnuts contain high amounts of ellagic acid related compounds, especially the ellagitannins pedunculagin, casuarictin, tellimagrandin I and II, glansrin A, B and C, stenophyllanin A, 2,3-hexahydroxydiphenoyl-b-D-glucopyranoside and rugosin, whereas almonds contain mainly procyanidins made up of catechin and epicatechin units [8]. Therefore, in terms of selecting a biomarker associated with nut intake, ellagitannin and procyanidin metabolites should be considered as biomarkers of consumption.

Different studies have shown that the addition of a moderate quantity of mixed nuts (30 g) to the daily diet contributes to the modification of plasma and urinary phenol metabolic profiles (Table 5). The most evident changes are related to the presence of urolithin A and B and their glucuronidated forms, after consumption of walnuts [8] and mixed nuts containing walnuts [78]. In addition, other ellagic acid colonic metabolites, such as isourolithin A, urolithin C and urolithin D, were also detected but at much lower levels [8,78]. As occurred with pomegranate, consumers of ellagitannin-rich nuts also showed different individual capacities for urolithin production [8]. Regarding ellagitannins, it is well documented that ellagic acid can be detected at very low amounts in biofluids, mainly in its dimethyl conjugated form after an acute intake or after medium length dietary interventions with 30 g peeled walnuts or with mixed nuts, respectively [8,78] (Table 5). In a similar study, also with metabolic syndrome patients and the same dietary intervention (30 g of mixed nuts), Mora-Cubillos et al. [79] showed that urolithin A glucuronide in plasma could be proposed as a clear biomarker of nut intake comparing with the control group.

Regarding almonds and hazelnuts, thirty metabolites belonged to the classes of hydroxyphenyl-γ-valerolactones, hydroxyphenylpropionic, hydroxyphenylacetic, hydroxycinnamic, hydroxybenzoic and hydroxyhippuric acids were associated with procyanidin (flavan-3-ols) metabolism (Table 5) [77,78]. However, a higher inter-individual variability was observed in the excretion of all metabolites derived from flavan-3-ols compared with urolithins. This suggests that ellagic acid metabolites (urolithins) could be proposed as nut intake biomarkers. However, when the contents of ellagic acid related compounds in nuts was low, it was not possible to detect urolithins in urine [77,78] suggesting the consideration of phenylvalerolactones as biomarkers of nut intake in parallel with urolithins.

### 4.2. Wine

Wine is consumed throughout the year in the Mediterranean countries where this is not constricted by religion. The phenolic composition of red wine has a particular combination of resveratrol, flavan-3-ols, procyanidins, anthocyanidins, ellagic acid related compounds and a broad spectrum of phenolic acids [65,80,81]. Biomarkers of red wine consumption have been studied both by monitoring the fate of an individual phenolic compound or through a broad analytical characterization of phenolic metabolites in plasma and urine [64,80,81,82,83,84]. The sustained intake of a moderate amount of red wine over 28 days in a healthy population produced an evident change in the phenol metabolic profile of their urine, plasma and faeces [3,80,81,83,84,85,86] (Table 6). After statistical analysis, including regression test of the data, a limited number of phenolic related compounds were proposed as good biomarkers of moderate red wine consumption. Among these metabolites were those derived from gallic acid, dihydroxyphenyl valerolactones, ethylgallate, epicatechin, phenolic acids and resveratrol. In addition, circulating hydroxytyrosol, the main biomarker of virgin olive oil consumption, was also detected in the plasma after red wine intake [87]. The presence of hydroxytyrosol in the plasma could be associated with wine ethanol intake via its interaction with the oxidative metabolism of dopamine [69].

Other phenolic compounds also proposed as biomarkers of wine consumption are anthocyanins, particularly malvidin-3-glucoside, which has been detected in both urine and plasma after an acute intake of red wine [88] (Table 6). Most of these phenol metabolites were also well correlated with wine intake in an epidemiological study in which participants gave a 24 h dietary recall [84]. Other studies have focused their analysis on resveratrol metabolites after the consumption of red wine. In this way, a significant increase of *trans*- and *cis*-resveratrol with different patterns of glucuronidation and sulphation as well as conjugated forms of the microbial metabolite dihydroresveratrol were observed in urine [86] (Table 6). These authors observed that the alcohol content of the wine did not increase the amount of resveratrol metabolites excreted in the urine [86].

## 5. Perspectives

Everything presented in this review leads us to deduce that, although it is difficult to correlate a specific phenol biological metabolite with the intake of a particular food, a selected group of these could be used as indicators of adherence to the MD. Table 7 summarizes the main phenol biological metabolites detected in the studies included in this review. Although each food or group of foods has its own specific phenolic composition, this particular characteristic is not usually reflected in the phenolic metabolome in biofluids. This is a consequence of the fast and intense metabolism that polyphenols undergo in the body. The pharmacokinetic behavior of phenolic metabolites in biofluids (plasma and urine) after an acute intake reveals two easily distinguished nutrikinetic patterns. The first kinetic corresponds to hepatic phase-II metabolism resulting in the presence of phase-II metabolites circulating in the plasma within 4 h of food ingestion, which are rapidly cleared by urinary excretion. These phase-II metabolites correspond to the original polyphenols present in the food which are absorbed in the upper intestinal tract (stomach and the different sections of the small intestine). The structural characteristics of these metabolites correspond to sulphate, glucuronide or methyl conjugates of the native phenols present in the food. The most important limitation for selecting these phenol phase-II metabolites as MD intake biomarkers is related to their short lifespan in circulation. This is explained by the character of xenobiotics which are rapidly metabolized and excreted in the urine, normally by 8 h post intake. Nevertheless, it is possible that the continuous exposure to a specific food through the diet, such as virgin olive oil or other non-seasonal plant-foods, could produce a “steady state”, maintaining a constant concentration in biofluids. There are some exceptions as in the case of anthocyanins, which have been detected as native glycosides in biofluids, the same form in which they are found in red grapes and red wine (Table 7). Similarly, free hydroxytyrosol has been detected after the intake of virgin olive oil or red wine. These native forms of phenol compounds in the foods could be good intake biomarkers of continuous exposure to a specific food in the context of the MD.

The second pharmacokinetic behaviour of phenol metabolites in biofluids is dominated by the appearance of colonic catabolites originated as a consequence of microbial fermentation in the large intestine. The substrates of this colonic metabolism are food phenols not absorbed during gastrointestinal digestion, in addition to those from enterohepatic recirculation (phase-II metabolites). These colonic metabolites are common to most MD foods and are represented by such simple phenolic acids as phenylacetic, phenylpropionic, benzoic acids and similar phenolic structures (Table 7). Their concentration in plasma produces a second peak (between 4 and 8 h after food intake) observed in many studies in which microbial metabolism was evaluated. The concentration of these circulating metabolites is usually higher than that corresponding to the first and fast phase-II metabolism. In addition to these simple and common phenolic acids, different microbial metabolites have been described for specific food phenolic compounds. Specifically, urolithins and phenyl-γ-valerolactones have been identified as microbial metabolites of ellagitannins and procyanidins after the intake of nuts or pomegranate products, between others (Table 7). These metabolites have a longer half-life, as was observed for phenolic acids. For example, the clearance of urolithins, which have been detected in urine 72 h after ingestion of ellagic acid derivatives, is slower than that of phase-II phenol metabolites.

Metabolites of mixed origin: indicates metabolites form endogenous (phase II metabolites) and those generated by gut microbiota. However, there are important challenges to be addressed. A major one is the need to find discriminated phenolic metabolites directly associated with the intake of a particular food or group of foods in order to offer solid candidates to be used as biomarkers of adherence to the MD. However, the intake of some foods is sporadic, normally because they are markedly seasonal, as is the case of some fruits or vegetables. Therefore, it should be adequate to consider a range biomarkers of food intake according to the period of the year when biofluids are analysed. It is evident that the same phenolic metabolite can be considered as intake biomarker of more than one MD food. On the other hand, other phenolic metabolites are exclusive to a one particular or a reduced number of foods. In this sense, short-term phenolic metabolites indicate recent consumption and can be used to validate some dietary assessment methods, such as 24 h recall of FFQ. Another challenge to highlight is the importance of unifying analytical methods and laboratory procedures to allow more consistence in the comparisons and data collected by different studies. So, a standardization of laboratory procedures including analytical instruments and protocols, sample collection and treatment, quantification and identification of phenolic compounds and their metabolites and homogeneity in the expression of results is necessary. One of the weak points is the lack of available commercial standards for identifying and quantifying phenolic biological metabolites. This is of special importance in the quantification of phase-II metabolites and specific colonic metabolites, such as urolithins or phenyl-γ-valerolactones.

Overall, three main groups of phenol biological metabolites can be proposed as good candidates for MD intake biomarkers. Firstly, there are the most food-specific phase-II metabolites with the limitation of their fast body clearance and the lack of standards for precise quantification in biofluids. Secondly, there are the colonic phenol metabolites represented by the simple phenolic acids (phenylacetic, phenylpropionic, benzoic acids and similar structures) and which are common to different plant-foods in the MD, together with the specific urolithins and phenyl-γ-valerolactones. Third are the native forms of polyphenols, for example anthocyanins or hydroxytyrosol, resulting from the sustained intake of certain foods over the year. However, an unmanageable aspect for the selection of phenol colonic metabolites as biomarkers of MD intake is the inter-individual variability observed in most human studies. These qualitative and mainly quantitative differences are mainly related to the variability in the microbiota composition defined as “metabotype” that result in differences in the phenol colonic catabolism.

The data presented in this review indicates that, instead of a single intake biomarker, a good proposal could be the study of the phenol metabolome in different populations of the Mediterranean area in order to link the concentration of specific phenol metabolites in biofluids, preferably in fasting urine, with the level of adherence to the MD (from low to high) based on 24 h recall of Food Frequency Questioners (FFQ).

## Figures and Tables

**Table 1 nutrients-13-03051-t001:** Phenolic metabolites (phase I, phase II) proposed as intake biomarkers of vegetables commonly consumed in the Mediterranean diet.

Food	Study Design (Type of Intake/Sampling)	Phenolic Metabolites	Location and Average Concentration (Tmax)	Ref.
Artichoke	Healthy subjects. Acute intake of 61.7 g steam cooked artichoke.	Ferulic acid (free + glucuronide)	P: 6.4 ng/mL (1–2 h); 8.4 ng/mL (>8 h)	[25]
	Phenolic dose: 487 mg chlorogenic acid derivatives and 11 mg apigenin and luteolin glycosides.	Dihydroferulic acid (free + glucuronide)	P: 21.8 ng/mL (>8 h)	
	Sampling: Fractionated blood (0–8 h).	Chlorogenic acid (free + glucuronide)	P: 6.4 ng/mL (1–2 h)	
		Caffeic acid (free + glucuronide)	P: 19.5 ng/mL (1–2 h)	
		3-(3′,4′-dihydoxyphenyl) propionic acid (free + glucuronide)	P: 12.1 ng/mL (>8 h)	
Artichoke	Healthy subjects. Acute intake of 3 capsules cont aining artichoke extract at 0, 4 and 8 h.	Ferulic acid (free + glucuronide)	U: 0.48–3.11 mg/24 h	[27]
	Phenolic dose (3 capsules): 124.2 mg caffeoylquinic acid derivatives and 19.8 mg luteolin glycosides.	Dihydroferulic acid (free + glucuronide)	U: 0.70–1.91 mg/24 h	
	Sampling: 24 h urine.	Isoferulic acid (free + glucuronide)	U: 0.17–0.63 mg/24 h	
		Vanillic acid (free + glucuronide)	U: 0.15–2.51 mg/24 h	
Onion	Healthy subjects. Acute intake of 47.5 g onion powder-enriched apple sauce.	Quercetin glucuronide	P: 213 ng/mL (<2.5 h)	[11]
	Phenolic dose: 180 μmol quercetin glycosides	Quercetin diglucuronide	P: 169 ng/mL (<2 h)	
	Sampling: Fractionated blood (0–24 h).	Methyl quercetin glucuronide	P: 90 ng/mL (<3 h)	
		Methyl quercetin diglucuronide	P: 65 ng/mL (<4 h)	
		Quercetin glucuronide sulphate	P: 43 ng/mL (<4 h)	
		Quercetin sulphate	P: 37 ng/mL (<2 h)	
Onion	Healthy subjects. Acute intake of 160 g stewed onions.	Quercetin glucuronide (sum of five types)	P: 2.3 μg/mL (<1 h)	[30]
	Phenolic dose: 331 μmol quercetin glucosides mainly, quercetin-3,4′-*O*-diglucoside and quercetin-4′-*O*-glucoside (equivalent to 100 mg quercetin).			
	Sampling: Fractionated blood (0–48 h).			
Onion	Healthy subjects. Acute intake of 270 g fried onions.	Quercetin-3′-sulphate	P: 605 nmol/L (<1 h)	[31]
	Phenolic dose: 275 μmol of total flavonoids from which 107 μmol were quercetin-3′,4′-diglucoside and 143 μmol to quercetin-4′-glucoside	Quercetin-3′-glucuronide	P: 351 nmol/L (<1 h); U: 1.8 μmol/24 h (0–8 h)	
	Sampling: Fractionated blood (0–24 h) and urine (0–24 h).	Quercetin glucuronide sulphate	P: 123 nmol/L (2.5 h); U: 2.7 μmol/24 h (0–8 h)	
		Methyl quercetin-3′-glucuronide	P: 112 nmol/L (<1 h); U: 1.8 μmol/24 h (0–8 h)	
		Quercetin diglucuronide	P: 62 nmol/L (<1 h); U: 2.2 μmol/24 h (0–8 h)	
Onion	Healthy volunteers. Acute intake of 200 g onion.	glucuronide sulphate, methyl quercetin glucuronide, methyl quercetin, quercetin glucoside sulphate, kaempferol glucuronide	Identification but not quantification of met abolites was done	[32]
	Phenolic dose: 53.8 mg quercetin.			
	Sampling: Fractionated urine (0–4 h).			
Spinach	Healthy volunteers. Acute intake 5 g ^13^C intrinsic ally labeled powdered spinach.	TMM-glucuronide	P: 19.1–54.3 nmol/L (>7 h)	[33]
	Phenolic dose: 160 μmol methoxyflavonols, form which 70 μmol TMM-4′-glucuronide.	TMM-sulphate	P: 38.7–125.5 nmol/L (>7 h)	
	Sampling: Fractionated blood (0–24 h).	Patuletin methyl glucuronide sulphate	P: nq	
		Spinacetin glucuronide sulphate	P: nq	
Tomato	Healthy subjects. Acute intake of 500 g tomato (T) or tomato sauce (TS)/70 kg bw	Naringenin	P: 15 nmol/L (1.4 h T); 22 nmol/L (0.5 h TS) U: 0.8 μmol/24 h (T); 2.1 μmol/24 h (TS)	[12]
	Phenolic dose: T 93537 ng/g homovanillic acid hexo side, 4788 ng/g naringenin, 2216 ng/g rutin, 1075 ng/g caffeoylquinic acid, 1572 ng/g caffeic acid hexo side, 3091 ng/g ferulic acid hexoside. Phenolic dose: TS: 138340 ng/g homovanillic acid hexoside, 7483 ng/g naringenin, 5389 ng/g rutin, 1075 ng/g caffeoylquinic acid, 1572 ng/g caffeic acid hexoside, 3091 ng/g ferulic acid hexoside.	Naringenin glucuronide	P: 53 nmol/L (2.1 h T); 131 nmol/L (0.8 h TS) U: 10.5 μmol/24 h (T); 59.5 μmol/24 h (TS)	
	Sampling: Fractionated blood (0–24 h) and urine (0–24 h).	Ferulic acid	P: nd; U: 7.9 μmol/24 h (T); 5.5 μmol/24 h (TS)	
		Isoferulic acid	P: 195 nmol/L (1 h T); 183 nmol/L (1 h TS) U: 100 μmol/24 h (T); 84 μmol/24 h (TS)	
		Ferulic acid glucuronide (sum of isomers)	P: 433 nmol/L (1–3 h T) 476 nmol/L (1.5–3 h TS) U: 121 μmol/24 h (T); 193 μmol/24 h (TS)	
		Ferulic acid sulphate (sum of isomers)	P: 156 nmol/L (1.4 h T) 279 nmol/L (1.5–2.3 h TS) U: 1437 μmol/24 h (T); 1295 μmol/24 h (TS)	
		Quercetin	P: 151 nmol/L (2 h T); 164 nmol/L (1.9 h TS) U: 0.6 μmol/24 h (T); 0.3 μmol/24 h (TS)	
		Quercetin glucuronide	P: nd U:0.4 μmol/24 h (T); 0.1 μmol/24 h (TS)	
		Quercetin sulphate	P: nd U:0.7 μmol/24 h (T); 0.2 μmol/24 h (TS)	
Tomato	Healthy subjects. Acute intake of 250 mL tomato sauce/70 kg bw.	Caffeic acid; caffeic acid glucuronide; caffeic acid sulphate;	P: nd; 6.4; 183 ng/mL U: 206; 55; 1396 ng/mL	[18]
	Phenolic dose: >1 μg/g homovanillic acid hexoside, 6.5 μg/g naringenin, 5.5 μg/g rutin, 2 μg/g caffeoylquinic acid, 2.3 ng/g caffeic acid hexoside, 3.5 μg/g ferulic acid hexoside.	5-caffeoylquinic acid	P: 2.7 ng/mL U: 17 ng/mL	
	Sampling: Blood (0 and 1 h) and urine (0–6 h).	Ferulic acid, ferulic acid glucuronide	P: 8.1; 54 ng/mL U: 453; 2852 ng/mL	
		Isoferulic acid	P: 95 ng/mL U: 22,569 ng/mL	
		Naringenin; naringenin glucuronide	P: 11.7; 73.4 ng/mL U: 39.1; 854 ng/mL	
		Quercetin; quercetin glucuronide; quercetin sulphate	P: 99; 21; 3.8 ng/mL U: 71; 14.4; 32 ng/mL	
		Dihydroxyphenyl propionic acid; dihydroxyphenyl propionic acid glucuronide; dihydroxyphenyl propionic acid sulphate	P: nd U: 195.; 2127; 2775 ng/mL	

P: plasma; U: urine; nq: not quantified; nd: not detected; TMM: 5,3′,4′-trihydroxy-3-methoxy-6,7-methylendioxyflavone.

**Table 2 nutrients-13-03051-t002:** Phenol metabolites (phase I, phase II) detected in human biofluids proposed as intake biomarkers of whole grains commonly consumed in the Mediterranean Diet.

Food	Study Design (Type of Intake/Sampling)	Phenolic Metabolites	Location and Average Concentration (Tmax)	Ref.
Whole grain wheat	Health subjects. Acute intake of 208 g of whole grain wheat bread.	3.5-dihydroxybenzoic acid (including glucuronic and sulphate)	U: 0.99 μmol/h (>8 h)	[36]
	Phenolic dose: 61 mg alkylresorcinols.	3,5-dihydroxyphenyl propionic acid (including glucuronic and sulphate)	U: 2.50 μmol/h (>7 h)	
	Sampling: Fractionated urine (0–24 h).	5-(3,5-dihydroxyphenyl)pentanoic acid (including glucuronic- and sulphate)	U: 0.20 μmol/h (>6 h)	
		3,5-dihydroxyhippuric acid	U: 0.24 μmol/h (>9 h)	
Whole rye, oat and wheat intake	Epidemiological study. 3-day food questionnaires recovery	Dihydroxybenzoic acid	U:18 μmol/24 h	[37]
	Average consumption of whole grains in the two visits was 72 g/day.	Dihydroxyphenyl propionic acid	U: 25 μmol/24 h	
	Sampling: 24 h urine. β-glucuronidase and sulphatase treatment.	Dihydroxycinnamic acid	U: 7 μmol/24 h	
		5-(3,5-dihydroxyphenyl)pentanoic acid	U: 0.9 μmol/24 h	
		Dihydroxycinnamic acid amide	U: 136 μmol/24 h	
		Dihydroxyhippuric acid	U: 4 μmol/24 h	
Whole grain wheat	Healthy overweigh/obese subjects. Replacement refined wheat with whole grain wheat for 8 weeks.	Ferulic acid	U: approximately 10 nmol/g creatinine; F: approximately 3000 nmol/kg S: approximately 3 nmol/L:	[38]
	Phenolic dose (daily): 96.7 mg ferulic acid, 26.5 mg sinapic acid, 9.4 mg coumaric acid.	Dihydroferulic acid	U: approximately 25 nmol/g creatinine; F: approximately 1500 nmol/kg	
	Sampling: Blood, urine, faeces.			

U: urine; S: serum; F: faeces; glu-: glucuronidated; sul-: sulphated.

**Table 3 nutrients-13-03051-t003:** Phenol metabolites (phase I, phase II) proposed as intake biomarkers of virgin olive oil.

Food	Study Design (Type of Intake/Sampling)	Phenolic Metabolites	Location and Average Concentration (Tmax)	Ref.
Phenol-enriched olive oil	Healthy subjects. Acute intake.	Hydroxytyrosol sulphate	P: 1.35 (1 h), 3.32 (1 h), and 4.09 µmol/L (1.5 h) after the intake of 250, 500 and 750 mg phenols/kg oil, respectively.	[4]
	Phenolic dose: 30 mL of phenol-enriched VOO contained 250, 500 and 750 mg phenolic compounds/kg oil.	Hydroxytyrosol acetate sulphate	P: 0.46 (1 h), 1.89 (2 h), and 2.24 µmol/L (1 h) after the intake of 250, 500 and 750 mg phenols/kg oil, respectively.	
	Sampling: Fractionated blood (0–6 h)	Homovanillic acid	P: 0.17 (1.5 h), 0.63 (1 h), and 0.65 µmol/L (1 h) after the intake of 250, 500 and 750 mg phenols/kg oil, respectively.	
		Homovanillic acid sulphate	P: 0.12 (1.5 h), 0.27 (1 h), and 0.53 µmol/L (1 h) after the intake of 250, 500 and 750 mg phenols/kg oil, respectively.	
Phenol-enriched olive oil	Hipercholesterolemic subjects. Sustained intake.	Phenylacetic acid	F: 39.46 µmol/L	[14]
	Phenolic dose: phenol-enriched (25 mL/day; 3 weeks).	2-(4-hydroxyphenyl)acetic acid,	F: 2.11 µmol/L	
	Sampling: Faeces	2-(3-hydroxyphenyl)acetic acid	F: 4.35 µmol/L	
		3-(4-hydroxyphenyl)propionic acid.	F: 1.99 µmol/L	
		Hydroxytyrosol	F: 2.21 µmol/L	
Virgin olive oil	Healthy subjects. Acute intake.	Methyl hydroxytyrosol + OH, met hyl oleuropein aglycone + OH, methyl DOA + OH, methyl oleuropein aglycone + H_2_O, methyl DOA + H_2_O, DOA + H_2_, DOA-H_2_, methyl DOA + OH, DOA + H_2_O, methyl DOA + H_2_O and DOA + CH_3_	U: 254 µg total Phase I and Phase II metabolites excreted (0–2 h)	[43]
	Phenolic dose: VOO 50 mL.			
	Sampling: Fractionated urine (0–6 h)			
Phenol-enriched olive oil	Hipercholesterolemic subjects. Sustained intake.	Hydroxytyrosol sulphate	P:1.35 μM (FVOO); 2.1 μM (FVOOT) U:14.6 μmol/24 h (FVOO); 9.4μmol/24 h (FVOOT)	[44]
	Phenolic dose: 25 mL/day virgin olive oil enriched with olive oil (FVOO) and thyme (FVOOT) phenolic compounds for 3 weeks. Daily serving FVOO: 500 mg/kg phenolic compounds mainly hydroxytyrosol derivatives (8.47 mg/day). FVOOT 500 mg phenolic compounds/kg mainly hydroxytyrosol derivatives (8.47 mg/day) and flavonoids (3 mg/day).	Hydroxytyrosol acetate sulphate	P:2.8 μM (FVOO); 2.3 μM (FVOOT) U: 17 μmol/24 h (FVOO); 7.8 μmol/24 h (FVOOT)	
	Sampling: Blood and 24 h urine	Homovanillic acid	P: 3.2 μM (FVOO); 1.9 μM (FVOOT) U: 12.8 μmol/24 h (FVOO); 11.4 μmol/24 h (FVOOT)	
		Homovanillic acid sulphate	P: 0.9 μmol/24 h (FVOO); 0.8 μmol/24 h (FVOOT) U: 23.5 μmol/24 h (FVOO); 23 μmol/24 h (FVOOT	
Virgin olive oil	Healthy subjects. Acute intake.	Hydroxytyrosol sulphate	P:5.2–6.7 ng/mL	[45]
	Phenolic dose: 40 mL of VOO. Phenolic content 400 mg phenols/L.	Monoglucuronide derivatives	P: 5.9–8.8 ng/mL	
	Sampling: Blood	Hydroxytyrosol	P: 2.6–3.9 ng/mL	
Virgin olive oil	Healthy adults. Acute intake.	3,4-DHPEA-EA + H_2_ + glucuronide and methyl 3,4-DHPEA-EA + H_2_ + glucuronide	U: 113 µg total Phase I and Phase II metabolites excreted (0–4 h)	[46]
	Phenolic dose: 50 mL VOO. Phenolic content 322 mg phenols/kg oil.			
	Sampling: Blood 0–6 h and urine 0–24 h			
Virgin olive oil	Healthy subjects. Acute intake.	Hydroxytyrosol glucuronide	nq	[47]
	Phenolic dose: 50 mL VOO.	Hydroxytyrosol sulphate	LDL cholesterol: 34.22 ng/mg ApoB	
	Sampling: Blood 0–1 h	Tyrosol glucuronide	LDL cholesterol: 0.96 ng/mg ApoB	
		Tyrosol sulphate	nq	
		Homovanillic acid sulphate	LDL cholesterol: 48.02 ng/mg ApoB	
Virgin olive oil	Hipercholesterolemic subjects. Sustained intake.	Hydroxytyrosol sulphate	HDL cholesterol: 49.48 ng/mg ApoB	[48]
	Phenolic dose: 25 mL/day VOO for 3 weeks (366 mg phenols/kg oil).	Homovanillic acid sulphate	HDL cholesterol: 18.30 ng/mg ApoB	
	Sampling: Blood	Homovanillic acid glucuronide	HDL cholesterol: 16.16 ng/mg ApoB	

FVOO: virgin olive oil enriched with its own phenolics; FVOOT: virgin olive oil enriched with its own phenolics plus thyme phenols; VOO: virgin olive oil. P: plasma; U: urine; F: faeces; DOA: deacetoxy oleuropein aglycone; *p*-HPEA-EDA and 3,4-HPEA-EDA: dialdehydic forms of deacetoxy of oleuropein and ligstroside aglycons, respectively; *p*-HPEA-EA and 3,4-DHPEA-EA: oleuropein and ligstroside aglycones-(3,4-dihydroxyphenyl)ethanol (3,4-DHPEA) and (*p*-hydroxyphenyl)ethanol (*p*-HPEA) linked to elenolic acid (EA); nq: not quantified.

**Table 4 nutrients-13-03051-t004:** Phenolic metabolites (phase I, phase II) proposed as intake biomarkers of the most representative ingested fruits in the Mediterranean diet.

Food	Study Design (Type of Intake/Sampling)	Phenolic Metabolites	Location and Concentration (Tmax)	Ref.
Apple juice	Healthy adults. Acute intake 1 L apple juice.	5-caffeoylquinic acid	P: 0.73 µmol/L (0.7 h); U:1.02 µmol/24 h	[50]
	Sampling: Fractionated blood (1–8 h) and 24 h urine. β-glucuronidase and sulphatase treatment	Caffeic acid	P: 0.09 µmol/L (2 h) U:0.11 µmol/24 h	
		4-*p*-coumaroylquinic acid	P: 0.09 µmol/L (1.3 h) U: 0.18 µmol/24 h	
		Phloretin	P: 0.17 µmol/L (2.1 h) U: 0.54 µmol/24 h	
		(−)-epicatechin	P: 0.05 µmol/L (0.9 h) U:0.29 µmol/24 h	
		Quercetin	P: 0.25 µmol/L (1.1 h) U: 0.10 µmol/24 h	
		Hippuric acid	U: 45.3 µmol/24 h	
		3-hydroxyphenyl propionic acid	U: 23.1 µmol/24 h	
		Dihydroxyphenyl propionic acid	U: 17.5 µmol/24 h	
		3- and 4-hydroxyhippuric acids	U: 17 and 13.6 µmol/24 h	
		3,4-dihydroxybenzoic and 4-hydroxybenzoic acids	U: 9.56 and 9.44 µmol/ 24 h	
		3-hydroxyphenyl acetic and 3,4-dihydroxyphenyl acetic acids	U: 6.17 and 7.07 µmol/24 h	
Cloudy apple juice	Healthy adults. Acute intake 250 mL cloudy apple juice	Phloretin glucuronide, naringenin glucuronide, (epi)catechin-methyl sulphate, vanillic acid sulphate, ferulic acid sulphate and feruloylquinic acid isomers (Tmax 1 h) Dihydroxyphenyl-γ-valerolactone glucuronide, catechol, hippuric, propionic and acetic acids (Tmax 5 h)	The metabolites were no quantified. Only changes in the chromatographic peak abundance of the main metabolites were studied.	[51]
	Sampling: Fractionated blood (0–5 h) and 24 h urine			
Apple fruit with peel	Healthy subjects. Acute intake apple quavers. 200 (low dose), 400 (medium dose) and 790 (high dose) g of apple quavers	Phloretin	U: 4 µg/mg creatinine (LD); 8 µg/mg creatinine (MD); 16 µg/mg creatinine (HD)	[52]
		Epicatechin (highest concentration after 3 h of intake),	U: 7 µg/mg creatinine (LD); 8 µg/mg creatinine (MD); 13 µg/mg creatinine (HD)	
	Sampling: Fractionated urine (0–24 h)	Procyanidin B2 (highest concentration after 3–6 h of intake)	U: 80 µg/mg creatinine (LD); 70 µg/mg creatinine (MD); 170 µg/mg creatinine (HD)	
Red-fleshed apple freeze-dried snack	Healthy subjects. Acute intake of 80 g of red-fleshed apple freeze-dried snack	Phloretin glucuronide	P: 61.0 nM; CB: 55 nM; U: 3.03 µmols/24 h (2–4 h)	[9,53]
	Phenolic dose: 42.3 mg anthocyanins, 88.0 mg phenolic acids, 13.8 mg flavanols, 17.3 mg flavonols, 0.42 mg flavanones, 33.7 mg dihydrochalcones.	Cyanidin-3-*O*-galactoside	P: 10.3 nM (2 h); U: 8.83 nmols/24 h (2–4 h)	
	Sampling: Fractionated blood, capillary blood and 24 h urine	Peonidin-3-*O*-galactoside	U: 3.65 nmols/24 h) (2–4 h)	
		Epicatechin sulphate	U: 1.63 µmols/24 h (2–4 h)	
		Epicatechin glucuronide	U: 1.12 µmols/24 h (2–4 h)	
		Epicatechin methyl-glucuronide conjugates	U: 1.01 µmols/24 h (2–4 h)	
		Dihydroxyphenyl propionic acid sulphate	CB: 120 nM (4 h)	
		Hydroxyphenyl-γ-valerolactone glucuronide	CB: 1300 nM (4 h	
		Catechol sulphate	CB: 2000 nM (12 h)	
		Catechol glucuronide	CB: 50 nM (12 h)	
		Methyl catechol sulphate	CB: 1500 nM (12 h)	
Pear	Male cyclists engaged in three 75 km cycling time trial.	Ferulic acid, dihydroferulic acid and 3-(4-hydroxyphenyl) propionic, hippuric acid and hydroxyhippuric acid	Metabolites were not quantified. Changes in the metabolic profile of plasma between pre and post-exercise were reporter.	[56]
	Phenolic dose: 0.15 g/kg carbohydrate from pears every 15 min until completing the 75-km time trial.			
	Sampling: Blood samples immediately and 1.5 h after 75 km time trial, and overnight fasted state at 21 h postexercise.			
Orange juice	Healthy men. Acute intake. Single dose of 1L or 0.5 L commercial orange juice.	Hesperetin (glucuronides and glucuronides-sulphates)	P: 1.25 μmol (>5 h, 1 L), 0.46 μmol (>5 h, 0.5 L) U: 15 μmol (>6 h, 0.5 L)	[57]
	Phenolic dose: 444 mg/L hesperetin, 96 mg/L naringenin.	Naringenin	P: 0.2 (>4 h, 1L) and 0.06 μmol (>4 h, 0.5 L) U: 5.9 μmol (0.5 L)	
	Sampling: Fractionated blood (0–24 h) and urine (0–48 h). β-glucuronidase and sulphatase treatment			
Orange	Healthy volunteers. Acute intake of 250 mL pulp enriched orange juice or placebo drink.	Hesperetin-*O*-diglucuronide	U: 9.9 μmol/24 h (2–5 h)	[58,59]
	Phenolic dose: 114 μmol Naringenin-7-*O*-rutinoside, 329 μmol hesperetin-7-*O*-rutinoside, 19 μmol 4′-*O*-methylnaringenin-7-*O*-rutinoside, 51 μmol naringenin-7-*O*-rutinoside-4′-*O*-glucoside, 19 μmol hesperetin-7-*O*-rutinoside-3′-*O*-glucoside, 5 μmol eriodictyol-7-*O*-rutinoside, 42 μmol apigenin-6,8-C-diglucoside and 5 μmol ferulic acid-4′-*O*-glucoside.	Hesperitin-*O*-sulphate-*O*-glucuronide and glucoside	U: 7.7 μmol/24 h (5–10 h) and 1.6 μmol/24 h (5–10 h)	
	Sampling: Fractionated urine (0–24 h).	Hesperitin-7 and 3-*O*-glucuronide	U: 4.7 μmol/24 h (2–5 h) and 19 μmol/24 h (5–10 h)	
		Hesperetin-3′-*O*-sulphate	U: 18.2 μmol/24 h (5–10 h)	
		Naringenin-*O*-diglucuronide	U: 2.5 μmol/24 h (5–10 h)	
		Naringenin-4′ and 7-*O*-glucuronide	U: 9.7 μmol/24 h (2–5 h) and 9.2 μmol/24 h (2–5 h)	
		Eriodictyol-*O*-sulphate	U: 0.26 μmol/24 h (2–5 h)	
		3-(3′hydroxy-4′-methoxyphenyl)hydracrylic acid	U: 43 μmol/24 h (5–10 h)	
		3-(3′hydroxyphenyl)hydracrylic acid	U: 17 μmol/24 h (10–24 h)	
		3-(3′hydroxy-4′-methoxyphenyl)propionic acid	U: 2.1 μmol/24 h (5–10 h)	
		Dihydroferulic acid	U: 5.2 μmol/24 h (5–10 h)	
		3′-methoxy-4′-hydroxyphenyl acetic acid	U: 3.9 μmol/24 h (10–24 h)	
		4′-hydroxyphenyl acetic acid	U: 22 μmol/24 h (10–24 h)	
		Hippuric acid	U: 317 μmol/24 h (10–24 h)	
		3-hydroxyhippuric acid	U: 0.5 μmol/24 h (5–10 h)	
Pomegranate	Healthy volunteers. Sustained intake 200 mL pomegranate juice for 3 weeks.	Urolithin A	F: 35.9 mg/g	[60]
	Phenolic dose 200 mL juice: 878.9 mg ellagic acid and ellagitannins, 41.5 phenolic acids, 38.0 anthocyanins, 3.39 flavonols, 1.19 flavan-3-ols.	Urolithin B	F: 9.47 mg/g	
	Sampling: faeces (lyophilized).	Urolithin C	F: 0.69 mg/g	
		Urolithin D	F: traces	
		Isourolithin A	F: 0.57 mg/g	
		Cyanidn-3-*O*-glucoside	F: 27.5 mg/g	
Pomegranate	Healthy volunteers. Acute intake. Pomegranate juice (PJ) and pomegranate extract (PE).	Ellagic acid	P: 0.06 μmol/L (0.65 h PJ) and 0.02 μmol/L (2.58 h PE)	[61]
	Phenolic dose: 857 mg gallic acid equivalent PJ and 776 mg gallic acid equivalents PE.	Urolithin A glucuronide	U: 1 μg/mL maximum concentration	
	Sampling: fractionated urine (0–24 h) and 24 h urine.			
Pomegranate	Healthy subjects. Acute intake. Pomegranate extract.	Urolithin A	Identified but not quantified metabolites	[62]
	Phenolic dose: 800 mg pomegranate extract: 330.4 mg punicalagins and 21.6 mg of EA.	Urolithin A glucuronide		
	Sampling: fractionated blood (0–24 h).	Urolithin B		
		Ellagic acid, methyl ellagic acid, dimethyl ellagic acid glucuronide	P: 33.8 ng/mL (1 h)	
Pomegranate	Patients with colon cancer diagnosis. Pomegranate extract (PE).	Urolithin A	P: 4.9 (PE-1); nd (PE-2) nM U: 2 (PE-1); 0.73 (PE-2) mg/g creatinine	[63]
	Phenolic dose: PE-1 low punicalagin:ellagic acid ratio; PE-2 high punicalagin:ellagic acid ratio.	Urolithin A glucuronide	P: 564 (PE-1); 124 (PE-2) nM U: 42 (PE-1); 7 (PE-2) mg/g creatinine	
	Sampling: Blood and urine.	Urolithin A sulphate	P: 39.9 (PE-1); 7.9 (PE-2) nM U: 0.6 (PE-1); 0.3 (PE-2) mg/g creatinine	
		Isourolithin A glucuronide	P: 564 (PE-1); 124 (PE-2) nM U: 8.5 (PE-1); ND (PE-2) mg/g creatinine	
		Isourolithin A	P: nq (PE-1); nd (PE-2) nM U: 0.7 (PE-1); ND (PE-2) mg/g creatinine	
		Urolithin B	P: 2.6 (PE-1); 10 (PE-2) nM U: 0.9 (PE-1); ND (PE-2) mg/g creatinine	
		Urolithin B glucuronide	P: 288.7 (PE-1); ND (PE-2) nM U: 19.4 (PE-1); ND (PE-2) mg/g creatinine	
		Urolithin B sulphate	P: 16.5 (PE-1); ND (PE-2) nM U: 0.05 (PE-1); ND (PE-2) mg/g creatinine	
		Urolithin C	P: nd U: 0.10 (PE-1); 0.01 (PE-2) mg/g creatinine	
		Urolithin D	P: 41.9 (PE-1); 19.5 (PE-2) nM U: nd	
		Ellagic acid, methyl ellagic acid, gallic acid	nd	
Red grape pomace drink	Healthy volunteers. Acute intake of 250 mL aqueous red grape pomace drink.	Benzoic acid-4-sulphate	P: 56.7 nM (3.0 h) U: 26 µmol/48 h	[6]
	Phenolic doset: 625 mg/100 mL total phenolic content.	Methylpyrogallol-sulphate	P: 512.4 nM (5.9 h) U: 93.6 µmol /48 h	
	Sampling: Fractionated blood (0–8 h and 24 h) and urine (0–48 h).	Protocatechuic acid-3-sulphate	P: 408.5 nM (2.1 h) U: 13.7 µmol/48 h	
		Gallic acid	P: 124.3 nM (3.8 h) U: 0.93 µmol/48 h	
		Vanillic acid-4-sulphate	P: 117 nM (4.0 h) U: 42.1 µmol/48 h	
		Ferulic acid 4-glucuronide	P: 72.8 nM (7.0 h) U: 5.1 µmol/48 h	
		Feruloylglycine	P: 26 nM (9.3 h) U: 12.4 µmol/48 h	
		(Epi)catechin-glucuronide	P: 135.5 nM (1.7 h) U: 1.7 µmol/48 h	
		(Epi)catechin-sulphate isomer 1	P: 87 nM (1.6 h) U: 2.6 µmol/48 h	
		(Epi)catechin-sulphate isomer 2	P: 94.9 nM (2.5 h) U: 2.4 µmol/48 h	
		5-(3′-Hydroxyphenyl)-γ-valerolactone-4′-glucuronide	P: 268.4 nM (5.3 h) U: 22.9 µmol/48 h	
		5-(4′-Hydroxyphenyl)-γ-valerolactone-3′-glucuronide	P: 1171 nM (5.2 h) U: 99.5 µmol/48 h	
		5-(Hydroxyphenyl)-γ-valerolactone-sulphate isomers	P: 893.7 nM (6.3 h) U: 205 µmol/48 h	
		5-Phenyl-γ-valerolactone-3′-glucuronide	P: 88.4 nM (9.1 h) U: 14.3 µmol/48 h	
		5-Phenyl-γ-valerolactone-3′-sulphate	P: 69.2 nM (11 h) U: 10.4 µmol/48 h	
Red grape juice	Healthy volunteers. Acute intake of 500 mL red grape juice.	Malvidin-3- *O*-glucoside	P: 120 nM (3 h) U: 22.4 μg (0–3 h), 27 μg (0–6 h)	[64]
	Phenolic dose (mg/mL): 233.6 malvidin-3-glucoside, 338.6 total anthocyanins, 64.3 flavan-3-ols, 3.7 resveratrol.			
	Sampling: Fractionated blood (0–6 h) and urine (0–6 h)			
Red grape juice	Healthy volunteers. Acute intake of 350 mL red grape juice.		Identification but not quantification	[65]
	Phenolic dose (µmol/L): main compounds 164.8 delphinidin-3-*O*-glucoside, 71.5 cyanidin-3-*O*-glucoside, 61.6 petunidin-3-*O*-glucoside; 50.7 delphinidin-3-*O*-(6″-*O*-*p*-coumaroyl)-5-*O*-diglucoside, 23 malvidin-3-*O*-glucoside, 50.9 gallic acid, 61.8 epicatequin, 20.4 catequin, 1.5 resveratrol, >100 procyanidins.	*O*-methyl-(epi)catechin-*O*-glucuronide and *O*-sulphate;	U	
	Sampling: Fractionated blood (0–24 h) and urine (0–24 h)	*O*-methyl-(epi)gallocatechin-*O*-sulphate	U	
		(−)-epicatechin-*O*-glucuronide and *O*-sulphate	U	
		3-*O*-glucosides of delphinidin and petunidin	U, P	
		3-*O*-glucosides of cyanidin, peonidin and malvidin	U	
		*O*-glucuronides of cyanidin, delphinidin, peonidin, petunidin and malvidin	U, P	
		3-(3′,4′-dihydroxyphenyl)propionic acid -3′-*O*-sulphate, caffeic acid-3′-*O*-sulphate, dihydrocoumaric acid, dihydroferulic acid-*O*-sulphate, ferulic acid-4′-*O*-sulphate, *p*-coumaric acid	U, P	
		3-(3′,4′-dihydroxyphenyl)propionic acid, coumaric acid-*O*-sulphate, dihydroferulic acid, caffeic acid-4′-*O*-sulphate, isoferulic acid-3′-*O*-sulphate and 3′-*O*-glucuronide	U	
		Caffeic acid and ferulic acid	P	
Red grape juice	Healthy volunteers. Acute intake of 350 mL red grape juice.	Delphinidin-3-*O*-glucoside	P: 1.4 nmol/L (1.4 h) U: 36 nmol/24 h	[66]
	Phenolic dose (µmol/350 mL): main compounds 58 delphinidin-3-*O*-glucoside, 25 cyanidin-3-*O*-glucoside, 22 petunidin-3-*O*-glucoside, 18 delphinidin-3-*O*-(6″-*O*-p-coumaroyl)-5-*O*-diglucoside, 8.1 malvidin-3-*O*-glucoside, 18 gallic acid, 22 epicatechin, 7.1 catechin, 0.5 resveratrol	Delphinidin-3-*O*-glucuronide	P: 1.5 nmol/L (3.3 h) U: 32 nmol/24 h	
	Sampling: Fractionated blood (0–24 h) and urine (0–24 h)	Petunidin-3-*O*-glucoside	P: 1.0 nmol/L (1.3 h) U: 17 nmol/24 h	
		Petunidin-3-*O*-glucuronide	P: 2.0 nmol/L (2.6 h) U: 368 nmol/24 h	
		Cyanidin-3-*O*-glucoside and *O*-glucuronide	P: uq U: 15 nmol/24 h and 19 nmol/24 h	
		Peonidin-3-*O*-glucoside and *O*-glucuronide	P: uq U: 5.7 nmol/24 h and 63 nmol/24 h	
		Malvidin-3-*O*-glucoside and *O*-glucuronide	P: uq U: 9.3 nmol/24 h and 46 nmol/24 h	
		(Epi)catechin-*O*-sulphate and *O*-glucuronide	P: ud U: 2301 nmol/24 h and 236 nmol/24 h	
		*O*-Methyl-(epi)catechin-*O*-sulphate and *O*-glucuronide	P: ud U: 1757 nmol/24 h and 45 nmol/24 h	
		*O*-Methyl-(epi)gallocatechin-*O*-sulphate	P: ud U: 137 nmol/24 h	
		*p*-Coumaric acid	P: 64 nmol/L (0.7 h) U: 0.5 nmol/24 h	
		*m*-Dihydrocoumaric acid	P: 355 nmol/L (5.8 h) U: 3 nmol/24 h	
		Dihydrocoumaric acid-*O*-sulphate	P: 27 nmol/L (6 h) U: 5.9 nmol/24 h	
		Caffeic acid	P: 178 nmol/L (0.5 h)	
		Caffeic acid-3′-*O*-sulphate	P: 47 nmol/L (1.0 h) U: 4.3 nmol/24 h	
		Dihydroxyphenyl propionic acid-3′-*O*-sulphate	P: 161 nmol/L (3.9 h) U: 17.1 nmol/24 h	
		Dihydroxyphenyl propionic-4′-*O*-sulphate	P: 42 nmol/L (4.4 h) U: 0.9 nmol/24 h	
		Ferulic acid	P: 63 nmol/L (1.8 h)	
		Ferulic acid-4′-*O*-sulphate	P: 63 nmol/L (1.2 h) U: 15.8 nmol/24 h	
Resveratrol and grape juice	Healthy volunteers. Acute intake of 0.03, 0.5 and 1 mg/kg resveratrol. 200, 400, 600 and 1200 mL red grape juice.	Resveratrol	U: 0.79 mg total excretion (dose 0.03 mg/kg)	[67]
	Phenolic dose (µmol/L): 7 resveratrol in grape juice	Resveratrol	U: 13.6 mg total excretion (dose 0.5 mg/kg)	
	Sampling: fractionated blood (0–5 h) and fractionated urine	Resveratrol	P: 0.75 mg/L (1.5 h); U: 15.4 mg total excretion (dose 1 mg/kg)	
		Resveratrol	U: not detected (dose 200 and 400 mL)	
		Resveratrol	U: <1% of ingested dose of resveratrol (dose 600 and 1200 mL)	
Red grape juice	Healthy volunteers. Acute intake of 400 mL red grape juice.	Cyanidin-3-*O*-glucoside	P: 0.42 ng/mL (0.5 h) U: 1.26 μg/h (0.5 h)	[69]
	Phenolic dose:283.5 mg anthocyanins, 15.2 mg flavan-3-ols, 5.6 mg flavanols, 9.2 mg resveratrol and 16.8 mg phenolic acids per single dose.	Delphinidin-3-*O*-glucoside	P: 6.12 ng/mL (0.5 h) U: 39.6 μg/h (0.5 h)	
	Sampling: Fractionated blood (0–3 h) and urine (0–7 h)	Malvidin-3-*O*-glucoside	P: 48.8 ng/mL (0.5 h) U: 86.7 μg/h (0.5 h)	
		Peonidin-3-*O*-glucoside	P: 27.3 ng/mL (0.5 h) U: 86.0 μg/h (0.5 h	
		Petunidin-3-*O*-glucoside	P: 16.1 ng/mL (0.5 h) U: 20.2 μg/h (0.5 h)	

P: plasma; U: urine; F: faeces; S: serum; CB: capillary blood; LD: low dose; MD: medium dose; HD: high dose; nq: not quantified; nd: not determined.

**Table 5 nutrients-13-03051-t005:** Phenolic metabolites (phase I, phase II) proposed as intake biomarkers of aromatic herbs and nuts commonly consumed in the Mediterranean Diet.

Food	Study Design (Type of Intake/Sampling)	Phenol Metabolites	Location and Average Concentration	Ref.
Oregano	Healthy subjects. Acute intake of 3.75 g oregano extract.	Caffeic acid	U: 29 μmol/48 h	[72]
	Phenolic dose: 47.7 mg rosmarinic acid, 0.56 gallic acid, 1.2 mg chlorogenic acid and 7.8 protocatechuic acid, 1.9 mg ferulic acid, 3.8 mg *p*-coumaric acid per serving.	Ferulic acid	U: 8 μmol/48 h	
	Sampling: 48 h urine. β-glucuronidase and sulphatase treatment	Syringic acid	U: 43 μmol/48 h	
		Vanillic acid	U: 95 μmol/48 h	
		*p*-hydroxybenzoic acid	U: 257 μmol/48 h	
		*p*-coumaric acid	U: 5 μmol/48 h	
		3,4-dihydroxyphenyl acetic acid	U: 31 μmol/48 h	
		m-hydroxyphenylacetic acid	U: 98 μmol/48 h	
Parsley	Healthy volunteers. Intake 20 g parsley for 7 days.	Apigenin	U: 20.7–5-27.3 μg/24 h	[74]
	Phenolic dose: 45 mg apigenin			
	Sampling: 24 h urine. β-glucuronidase and sulphatase treatment.			
Walnuts	Subjects diagnosed benignant prostate hyperp lasia or prostate cancer. Intake of 35 g peeled walnuts for 3 days.	Urolithin A glucuronide Dimethyl ellagic acid	P: 0.11 μM (high excreters); U: >5 μM (high excreters), <5 μM (low excreters), absence (very low excreters) PT: 0.5–2 ng/g tissue	[8]
	Phenolic dose: 202 mg ellagitannins and 8 mg of free ellagic acid	Urolithin B	P: nq U: nq PT: nq	
	Sampling: Prostate tissue (PT), blood and urine.	Urolithin C	P: nq U: nq PT: nq	
		Urolithin C methyl ether glucuronide	P: nq U: nq	
Mixed nuts	Healthy volunteers. 30 g nuts (15 g walnuts, 7.5 g hazelnuts and 7.5 g almonds) for 3 days.	3,4-dihydroxyphenyl valerolactone (glucuronide, sulphate, and sulphoglucuronide)	U: 15,500 μg/g creatinine	[76]
	Phenolic dose: 63 mg ellagic acid equivalents and 42 mg proanthocyanidins.	3-hydroxyphenylpropio-2-ol	U: 200 μg/g creatinine	
	Sampling: Urine.	4-hydroxyphenylacetic acid	U: 2300 μg/g creatinine	
		4-hydroxybenzoic acid	U: 1700 μg/g creatinine	
Mixed nuts	Subjects with metabolic syndrome. Intake of 30 g of mixed nuts (15 g walnuts, 7.5 g hazelnut and 7.5 g almonds) daily during 12 weeks.	Ellagic acid	U: 3 μmol/24 h increment	[77]
	Phenolic dose: 37.9 mg ellagic acid equivalents and 62.4 mg proanthocyanidins per serving.	Urolithin A (mainly glucur onide)	U: 50 μmol/24 h increment	
	Sampling: 24 h urine.	Urolithin B (mainly glucuronide)	U: 23 μmol/24 h increment	
		5-(dihydroxyphenyl)-γ-valerolactone	U: 133 μmol/24 h increment	
		Mono and di methylated ellagic acid, Urolithin C and D	U: nq	

U: urine, P: plasma; PT: prostate tissue; nq: not quantified.

**Table 6 nutrients-13-03051-t006:** Phenolic metabolites (phase I, phase II and colonic) proposed as intake biomarkers of red wine consumption in the Mediterranean Diet.

**Food**	**Study Design (Type of Intake/Sampling)**	**Phenolic Metabolites**	**Location and Average Concentration (Tmax)**	**Ref.**
Red wine	Healthy subjects. Sustained intake of 200 mL/day of RW 4 weeks.	*trans*-resveratrol-3-*O*-glucuronide	U: 500 nmol/g creatinine	[3]
	Phenolic dose; Total resveratrol 2.56 mg/200 mL red wine and 20 g/day alcohol	*cis*-resveratrol-3-*O*-glucuronide	U: 175 nmol/g creatinine	
	Sampling: Blood and urine			
Dealcoholized red wine	Healthy subjects. Acute intake of 100 mL dealcoholized red wine.	Malvidin glucoside	P: 7.01 nM (Cmax) U: 0.06 µmol/24 h	[79]
	Phenolic dose: (100 mL wine): anthocyanins (22.1 mg), phenolic acids (22.2 mg), procyanidins (6.60 mg), flavonols (20.5 mg), stilbenes (3.21 mg), phenyl alcohols (tyrosol and hydroxytyrosol, 2.23 mg).	Gallic acid sulphate	P: 76.8 nM (Cmax) U: 2.42 µmol/24	
	Sampling: fractionated blood (0–6 h), fractionated urine (0–24 h).	Gallic acid glucuronide	U: 0.43 µmol/24 h	
		Syringic acid sulphate	P: 159 nM (Cmax) U: 3.10 µmol/24 h	
		Syringic acid glucuronide	P: 10.1nM (Cmax) U: 3.25 µmol/24 h	
		Caffeic acid sulphate	P: 60.3 nM (Cmax)	
		Ferulic acid sulphate	P: 23.2 nM (Cmax)	
		Ferulic acid glucuronide	U: 2.28 µmol/24 h	
		Protocatechuic acid sulphate	U: 3.31 µmol/24 h	
		Dihydroxyphenyl acetic acid	U: 21.1 µmol/24 h	
		Resveratrol sulphate	P: 410 nM (Cmax) U: 30.3 µmol/24 h	
		Resveratrol glucuronide	P: 3.09 nM (Cmax) U: 2.63 µmol/24 h	
		Catechin sulphate	P: 62.5 nM (Cmax) U: 10.9 µmol/24 h	
		Catechin glucuronide	U: 0.22 µmol/24 h	
		Epicatechin sulphate	P: 63.4 nM (Cmax) U: 1.65 µmol/24 h	
		Epicatechin glucuronide	P: 45.9 nM (Cmax) U: 1.23 µmol/24 h	
		Methyl catechin sulphate	P: 17.9 nM (Cmax) U: 1.46 µmol/24 h	
		Methyl epicatechin sulphate	P: 24.1 nM (Cmax) U: 2.55 µmol/24 h	
		Methyl catechin glucuronide	P: 16.7 nM (Cmax) U: 0.60 µmol/24 h	
		Methyl epicatechin glucuronide	P: 8.3 nM (Cmax) U: 0.79 µmol/24 h	
		Dihydroxyphenyl-γ-valerolactone	U: 31.5 µmol/24 h	
		Hydroxytyrosol	P: 45.9 nM (Cmax) U: 2.17 µmol/24 h	
		Hydroxytyrosol sulphate	P: 333 nM (Cmax) U: 7.48 µmol/24 h	
Red wine	Healthy subjects. Sustained intake of 250 mL red wine for 4 weeks.	3,5-dihydroxybenzoic acid	F: 0.35 μg/g	[80]
	Phenolic dose: 1758 mg gallic acid Eq/L, 447 mg malvidin-3-*O*-glucoside/L and 1612 mg (+)-catechin/L.	Protocatechuic acid	F: 1.25 μg/g	
	Sampling: Faeces.	Vanillic acid	F: 1.12 μg/g	
		3-hydroxyphenyl acetic acid	F: 18.60 μg/g	
		Syringic acid	F: 1.84 μg/g	
		4-hydroxy-5-(3′,4′-dihydroxyphenyl)valeric acid	F: 1.65 μg/g	
		4-hydroxy-5-phenylvaleric acid	F: 241 μg/g	
		5-(3′-hydroxyphenyl)-γ-valerolactone	F: 24.3 μg/g	
Red wine	Healthy subjects. Acute intake of 5 mL red wine/kg bw.	Caffeic acid	P: 84 nmol/L (2 h)	[81]
	Phenolic dose: caffeic acid 11 mg/L, protocatechuic acid 1.58 mg/L and 9.5 mg/L gallic acid.	4-*O*-methylgallic acid	P: 176 nmol/L (2 h)	
	Sampling: Fractionated blood (0–4 h).			
Red wine	Healthy volunteers. Sustained intake of 272 mL/day red wine for 4 weeks.	Gallic acid metabolites	P: 0.13 μmol/L U: 35 μmol/24 h	[82]
	Sampling: Blood and 24 h urine	Dihydroxypheyl valerolactone	P: 1.07 μmol/L U: 1083 μmol/24 h	
		Methylgallate metabolites	P: 0.51 μmol/L U: 576 μmol/24 h	
		Epicatechin metabolites	P: 0.08 μmol/L U: 76 μmol/24 h	
		Resveratrol	U: 5352 μmol/24 h	
		Resveratrol microbial metabolites	U: 4208 μmol/24 h	
Red wine (RW) Dealcoholized red wine (DRW)	Cardiovascular risk patients. Sustained intake of 272 mL of red wine and dealcoholized red wine during 4 weeks.	*trans*-resveratrol-4 and 3-glucuronide	U: 838 (DRW); 391 (RW) and 193 (DRW); 193 (RW) nmol/24 h	[85]
	Sampling: 24 h urine	*cis*-resveratrol-4 and 3-glucuronide	U: 193 (DRW); 193 (RW) nmol/24 h	
		*cis*-resveratrol-3-glucuronide	U: 487 (DRW); 450 (RW) and 2410 (DRW); 2305 (RW) nmol/24 h	
		Transresveratrol-4 and 3-sulphate	U: 141 (DRW); 91 (RW) nmol/24 h	
		*cis*-resveratrol-4 and 3-sulphate	U: 489 (DRW); 592 (RW) and 1045 (DRW); 932 (RW) nmol/24 h	
		*trans*-resveratrol-3,4-disulphate	U: 891 (DRW); 753 (RW) and 414 (DRW); 419 (RW) nmol/24 h	
		Resveratrol sulphate glucuronide	U: 211 (DRW); 170 (RW) nmol/24 h	
		*trans*-piceid	U: 2.94 (DRW); 2.63 (RW)nmol/24 h	
		*cis*-piceid	U: 14.7 (DRW); 17.8 (RW) nmol/24 h	
		Piceid glucuronide	U: 29.6 (DRW); 31.8 (RW) nmol/24 h	
		Piceid sulphate	U: 95.3 (DRW); 94.1 (RW) nmol/24 h	
		Dihydroresveratrol	U: 18.1 (DRW); 20.3 (RW) nmol/24 h	
		Dihydroresveratrol glucuronide (sum of 2 isomers)	U: 593 (DRW); 529 (RW) nmol/24 h	
		Dihydroresveratrol sulphate (sum of 2 isomers)	U: 4125 (DRW); 3480 (RW) nmol/24 h	
		Dihydroresveratrol sulphate glucuronide	U: 333 (DRW); 300 (RW) nmol/24 h	
Red wine	Healthy subjects. Acute intake of a single dose 400 mL red wine.	Cyanidin-3-*O*-glucoside	P: n.d U: 0.66 μg/h (2.5 h)	[69]
	Phenolic dose:279.6 mg anthocyanins, 74.8 mg flavan-3-ols, 2.4 mg flavanols, 6.8 mg resveratrol and 23.2 mg pheolic acid in 400 mL.	Delphinidin-3-*O*-glucoside	P: n.d U: 14.9 μg/h (0.5 h)	
	Sampling: Fractionated blood (0–3 h) and urine.	Malvidin-3-*O*-glucoside	P: 18.5 ng/mL (1.5 h) U: 60.2 μg/h (1.5 h)	
		Peonidin-3-*O*-glucoside	P: 12.6 ng/mL (1.5 h) U: 44.1 μg/h (0.5 h)	
		Petunidin-3-*O*-glucoside	P: 12.6 ng/mL (1.5 h) U: 20.5 μg/h (1.5 h)	

P: plasma; U: urine; F: faeces.

**Table 7 nutrients-13-03051-t007:** Summary of the phenol biological metabolites proposed as a useful tool for the assessment of Mediterranean Diet adherence.

Phase II Metabolites of Native Phenols	F	G	S	M	GS	MG	MS	AS	Precursor	Food
Caffeoyl quinic acid	p,u	p							Caffeic acid and derivatives	Apple/artichoke/tomato
Catechin		u	u			p,u	p,u		Catechin and derivatives	Wine
Cyanidin-3-*O*-galactoside	p,u								Cyanidin	Apple
Cyanidin-3-*O*-glucoside	p,u,f	p,u							Cyanidin	Pomegranate/Red grape/wine
Delphinidin-3-*O*-glucoside	p,u	p,u							Delphinidin	Red grape
Dihydroresveratrol	u	u	u		u				Resveratrol	Wine
Epicatechin		p,u	p,u			p,u	p,u		Epicatechin/Procyanidins	Apple/red grapes/wine
Eriodictyol			u						Eriodictiol/Naringenin	Orange
Hesperetin		p,u	p,u						Hesperetin	Orange
Hydroxytyrosol	p,u,f		p,u	u				p, u	Hydroxytyrosol, oleuropein derivatives	Virgin olive oil/wine
Kaempferol		u							Kaempferol	Onion
Malvidin-3-*O*-glucoside	p,u	p,u							Malvidin	Red grape/wine
Methyl patuletin					p				Patulein	Spinach
Naringenin	p,u	p,u							Naringenin	Orange/tomato
Oleuropein derivatives	u	u		u					Oleuropein	Olive oil
Peonidin-3-*O*-galactoside	p								Penonidin	Apple
Peonidin-3-*O*-glucoside	p,u	p,u							Peonidin	Red grape
Petunidin-3-*O*-glucoside	p,u	p,u							Petunidin	Red grape
Quercetin	p,u	p,u	p	u	p,u	p,u			Quercetin	Onion/tomato
Spinacetin					p				Spinacetin	Spinach
Phloretin	p,u	p,u							Phloretin	Apple
Piceid	u	u	u						Piceid	Wine
Resveratrol		p,u	p,u		p,u				Resveratrol	Wine
TMM		p	p						TMM	Spinach
Tyrosol									Tyrosol/Ligstroside derivatives	Virgin olive oil
Di and Hydroxyphenyl acetic acid	p,u,f								Phenolic acids/flavan-3-ols/procyanidins/flavanones	Apple /nuts/orange/oregano/wine
Hydroxyphenyl acetic acid methoxy	u								Hesperetin/Naringenin	Orange
Hydroxyphenylpropionic acid	u,f								Phenolic acids/flavan-3-ols/procyanidins/flavanones	Apple/pear
Dihydroxyphenylpropionic acid	p,u	p	p,u						Caffeic and ferulic acids/epicatechin/procyanidins	Apple/artichoke/red grapes/olive oil/tomato/whole grains
Dihyroxyphenyl propionic acid amide									Alkylresorcinols	Whole grains
Dihydroxymethoxyphenyl propionic acid	u								Hesperitin/Naringenin	Orange
Dihydroxyphenoyl hydracrylic acid	u								Hesperitin/Naringenin	Orange
Dihydroxymethoxyphenyl hydracrylic acid	u								Hesperitin/Naringenin	Orange
Dihydroxy benzoic glycine									Alkylresorcinols	Whole grains
Pyrogallol							p,u		Gallic acid	Red grapes
Hydroxyphenyl valeric acid	f								Epicatechin/Procyanidins	Wine
Dihydroxyphenyl valeric acid	f								Epicatechin/procyanidins	Wine
5-(3,5-dihydroxyphenyl)pentanoic acid	u	u	u						Alkylresorcinols	Whole grains
Phenyl-γ-valerolactone		p,u	p,u						Epicatechin/Procyanidins	Apple/pomegranate/red grapes/wine/nuts
Hydroxyphenyl-γ-valerolactone	f	p,u	p,u						Epicatechin/Procyanidins	Apple/pomegranate/red grapes/wine/nuts
Dihydroxyphenyl-γ-valerolactone		p,u	p,u		u				Epicatechin/Procyanidins	Apple/pomegranate/red grapes/wine/nuts
Urolithin A and B	p,u,f	p,u	p,u						Ellagitannins/ellagic acid	Nuts/pomegranate
Urolithin C, urolithin D	p,f								Ellagitannins/ellagic acid	Nuts/pomegranate
Isourolithin A	p,u	p,u							Ellagitannins/ellagic acid	Pomegranate
**Metabolites of Mixed Origin**	**F**	**G**	**S**	**M**	**GS**	**MG**	**MS**	**AS**	**Precursor**	**Food**
Caffeic acid	p,u	p,u	p,u						Caffeic and ferulic acids and derivatives	Apple/artichoke/oregano/red grapes/tomato/wine
Catechol	p,u	p	p				p		Catechol derivatives/microbial derivatives	Apple/Wine
Coumaric acid	p,u		u						Coumaric acid derivatives	Oregano/red grapes
Dihydroxybenzoic acid	u,f	u	u						Benzoic acid derivatives/microbial derivatives	Apple/nuts/whole grains
Dihydroferulic acid	p,u,f	p,u	p,u						Caffeic and ferulic acids and derivatives	Artichoke/pear
Ellagic acid	p			p		p			Ellagic acid/ellagitannins	Nuts/pomegranate
Ferulic acid	p,u,f	p,u	p,u						Caffeic and ferulic acids and derivatives	Apple/artichoke/nuts/oregano/pear/red grape/tomato/grains/wine
Gallic acid	p,u	u	p,u	p					Gallic acid	Red grapes/wine
Homovanillic acid	p,u		p,u						Vanillic acid derivatives/microbial derivatives	Virgin olive oil
Hydroxybenzoic acid	p,u								Benzoic acid derivatives/microbial derivatives	Apple/nuts/orange/oregano
Isoferulic acid	p,u	u	u						Caffeic and ferulic acids and derivatives	Artichoke/red grapes/tomato
Protocatechuic acid			p,u						Cyanidin	Red grapes/wine
Syringic acid		u	u						Malvidin/phenolic acids	Oregano/wine
Vanillic acid	u,f	u	p,u						Vanillic acid derivatives/microbial derivatives	Apple/artichoke/oregano/pear/red grapes/wine

F: free; G: glucuronide; S: sulphate; M: methyl; GS: glucuronide-sulphate; MG: methyl glucuronide; MS: methyl sulphate; MGS: methyl-glucuronide-sulphate; AS: acetyl sulphate. TMM: 5,3′,4′-trihydroxy-3-methoxy-6,7-methylendioxyflavone. p: plasma, u: urine; f: faeces. Human biofluids in which metabolites were detected.

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
