# Peer review of "Phenol Biological Metabolites as Food Intake Biomarkers, a Pending Signature for a Complete Understanding of the Beneficial Effects of the Mediterranean Diet"

_nutrients, 2021, doi:10.3390/nu13093051_

Round 1

Reviewer 1 Report

This paper presents an excellent review of potential biomarkers for adherence to a Mediterranean Dietary pattern. It is fine as is, with a few grammatical/spelling suggestions below:

Line 24 – dietary interventionS

Line 41 – considerING them

Line 46 – diverse typeS of foods

Line 57 -  most of food components

Line 187 – its most representative polyphenols

Line 191 – behaves in a biphasic way

Author Response

Attached document

Reviewer 2 Report

This is a very interesting paper, and I congratulate the authors for all the hard work that has been done. However, more work is required to make it a paper that will be utilised as a resource and referred to by future papers.

General:

A rationale needs to be provided as to how foods were selected for this paper, as you have not included all components of the traditional MedDiet. Many, many plant foods (including many more herbs and spices) have not been included. Green leafy vegetables are a key feature of the diet and you have only included spinach – we need inclusion of all the “horta” (wild greens) that are consumed throughout the Mediterranean region.

It would also help if the written section for each of the foods was the same, that is name (perhaps scientific name, varieties), what known compounds are in them. List the cellular studies, then animal studies and then human studies. You could then finish off with a summary of the pharmacokinetics (is it a linear relationship, what is the half-life).

Could you please identify if a systematic search strategy was used at all? If not how did you find the articles? This will tie in with the rationale of selecting foods.

This paper, perhaps in a summary table, needs a description of the health benefits of each of the group of phenols. There is no mention of benefits throughout the paper.

Line 162 – some discussion of the problems associated with current methods of assessing dietary intake. Is it expected that the metabolomics will be more accurate and cheaper to run than for example an FFQ? You can also compare blood vs urine – as there is clear difference in the level of invasiveness to the participant.  Are there are any papers that have assessed accuracy in dietary assessment techniques with any biomarker. This will add to your rationale for why this paper is important.

Table 1  (and for all of the food tables)

  • It would be more logical to have study design in the column just after food…so it would read..
  • Food>>Study Design>>Phenolic metabolite>>Location and concentration (this is relevant for all of the tables)
  • Could you please indicate if there was an intake of any other foods in the time period of sampling urine or plasma
  • (study no 30 in table) For quercetin could you indicate what the five types are? Where the onions stewed with any liquid – this may impact loss of phenols
  • Is there anyway to get the same expression of units for the same phenols in the same food across studies (ie in Table 1 - study 11, 30 and 31) and (12 & 18)

Line 203 – could you let us know what the phenols in lettuce are, despite there being no studies. Do they differ for different lettuces?

Line 221 – you report there is high variability between participants, I’m reading this as high variability between participants with the same intake(?)– if my assumption is correct could you indicate the degree of variability. If my assumption is incorrect could you please reword for clarity.

Line 224 - Is the correlation seen btw urinary levels and intake linear, or is there upper and lower thresholds?

How did you select which vegetables to include in this review – I assume as you present limited number of vegetables, that if there were studies that you included them. if this is the case why did you report lettuce.  

Line 289 – there will be confusion over why you have referred to VOO as opposed to EVOO – can your rationale. Is VOO actually EVOO? If they are different you will need to report on EVOO as well. For example you have not mentioned the phenol, oleocanthal.

Line 407 – please provide a reference for your statement that red apples the most frequently eaten fruit in the MD.

Table 4 – I appreciate why you have included juices in this, but there needs to be commentary on whether juices were consumed in the MEDDiet and whether they are recommended as part of a healthy diet?

Line 463 – I agree pears are delicious, as are all the other fruits. This is not an appropriate statement for a scientific article.

Line 480 – could you provide a reference that oranges reach their maximum nutritional status in autumn and winter, and please state which nutrients you are referring to here.

Line 597 – please review grammar.

Line 598 – review grammar, and please include the end of the brackets in the sentence.

Line 677 – please review sentence, this review should provide evidence for your finding, stating “think” makes it seems as you have not deduced this from the evidence base.

Line 694 – should ‘their’ be ‘the’

Table 7 needs to also include in which fraction of the body these metabolites have been found.

Author Response

Attached document
